# The role of explicit memory in syntactic persistence: Effects of lexical cueing and load on sentence memory and sentence production

**Chi Zhang**[1]*, **Sarah Bernolet**[2], **Robert J. Hartsuiker**[1]

**1** Department of Experimental Psychology, Ghent University, Ghent, Belgium, **2** Department of Linguistics, University of Antwerp, Antwerp, Belgium

* chi.zhang@ugent.be

## Abstract

Speakers' memory of sentence structure can persist and modulate the syntactic choices of subsequent utterances (i.e., structural priming). Much research on structural priming posited a multifactorial account by which an implicit learning process and a process related to explicit memory jointly contribute to the priming effect. Here, we tested two predictions from that account: (1) that lexical repetition facilitates the retrieval of sentence structures from memory; (2) that priming is partly driven by a short-term explicit memory mechanism with limited resources. In two pairs of structural priming and sentence structure memory experiments, we examined the effects of structural priming and its modulation by lexical repetition as a function of cognitive load in native Dutch speakers. Cognitive load was manipulated by interspersing the prime and target trials with easy or difficult mathematical problems. Lexical repetition boosted both structural priming (Experiments 1a–2a) and memory for sentence structure (Experiments 1b–2b) and did so with a comparable magnitude. In Experiment 1, there were no load effects, but in Experiment 2, with a stronger manipulation of load, both the priming and memory effects were reduced with a larger cognitive load. The findings support an explicit memory mechanism in structural priming that is cue-dependent and attention-demanding, consistent with a multifactorial account of structural priming.

## Introduction

A central issue of language production is the role of working memory in the production processes. Speaking often proceeds at a fast rate, suggestive of a view by which speakers do not always need working memory to control the formulation of utterances. In fact, in his classic book *Speaking*, Levelt [1] argued that other than in determining the message that speakers intend to utter, most of the processes involved in speaking function "in a reflex-like, automatic way". However, speakers do sometimes show limited memory capacity in stages later than message planning. For example, the number of words speakers are able to plan in advance is

**Data Availability Statement:** All the relevant data and scripts are available from the Open Science Framework (DOI https://doi.org/10.17605/OSF.IO/6UTKF).

**Funding:** CZ was supported by China Scholarship Council (No. 201606280023). The funders had no role in study design, data collection and analysis, decision to publish, or preparation of the manuscript.

**Competing interests:** The authors have declared that no competing interests exist.

reduced when they are distracted by a secondary task [2]. Such evidence suggests that speakers might store a certain number of lexical items in their short-term memory before their utterance starts and that they have to assign attentional resources to maintain these memory traces. Important for our purposes, some accounts of syntactic encoding argued that memory retrieval also facilitates the subsequent syntactic choice (e.g., [3]). Here we tested two predictions of that account: (1) that speakers make use of lexical cues to reinforce an effect of memory retrieval on syntactic encoding; (2) that such a memory effect is constrained by the limited memory capacity of speakers.

Grammatical encoding is the process in which speakers map the concepts to ordered sequences of words that feature grammatical functions (e.g., subject, object). Sentence production models often envision grammatical encoding as an automatic process that requires little involvement of working memory (e.g., [1, 3]). However, recent evidence has demonstrated that certain aspects of syntactic processing in sentence production are constrained by limited memory capacity (e.g., [2, 4–6]). Because only a limited amount of information can be kept in working memory [7], a process can be shown to require working memory if a concurrent task that also occupies working memory capacity interferes with this process. This phenomenon is often referred to as the cognitive load effect. Such load effects have been found in grammatical encoding processes of sentence production. For example, if speakers formulated sentences when their working memory was occupied by a word list they had to remember, the accessibility of the previously memorized lexical information was reduced, resulting in a change of word-order preference [6]. Similarly, speakers produced more subject-verb agreement number errors in sentence production when there was a cognitive load [5]. Hartsuiker and Moors [8] argued against a strictly dichotomous view of automatic vs. non-automatic processes and suggested a gradual view by which processes like syntactic processing in sentence production have some but not all "automaticity features" (such as being constrained by working memory).

This non-binary view of automaticity in syntactic processing meshes nicely with studies on structural priming, which also show both automatic and non-automatic effects. These studies tap into the persistence of syntactic structure, in particular how recently experienced sentence structures influence the subsequent syntactic choice ([9, 10]). Structural priming as an experimental paradigm is advantageous in many ways for psycholinguists to understand the comprehension, production, and acquisition mechanisms that involve syntactic representations [11]. However, there is still debate about the underlying mechanisms of structural priming. Some accounts posited that the syntactic structure persists automatically (e.g., [12]), whereas others argued that some processes in structural priming are constrained by limited memory capacity (e.g., [13, 14]). In particular, the latter account proposes that structural priming is driven by an automatic process of implicit learning and a non-automatic, explicit memory-related process. The explicit memory-related process would involve retrieval and adaptation of the previous sentence, and such retrieval would be more likely to occur if words are repeated between prime and target sentences (lexical cueing effect). The goal of the current study is to pinpoint the non-automatic (i.e., explicit memory-related) components in syntactic persistence. We asked whether lexical overlap elicits cueing effects on the retrieval of sentence structure from memory and whether the effects of structural priming varies with the limited capacity of memory.

Bock [9] first discovered structural priming in an experimental setting: In a series of tasks disguised as a recognition memory test, speakers were more likely to choose a passive structure (e.g., *The church is being struck by lightning*) over an active structure (e.g., *The lightning is striking the church*) to describe a picture after they heard a passive sentence. Structural priming is often considered as an effect that entails the autonomous repetition of syntactic structures independent of processing at other linguistic levels such as lexical access ([9, 15, 16], but see

[17]). Nevertheless, it was also found that lexical overlap considerably enhances the magnitude of structural priming (i.e., lexical boost). Pickering and Branigan [12] demonstrated that, in a sentence completion task, the chance for a speaker to complete a target sentence fragment (e.g., *The patient showed. . .*) with an argument structure of [-NP-NP] increased after completing prime fragments like *The racing driver showed the helpful mechanic. . .* or *The racing driver gave the helpful mechanic. . .* because the speaker was forced to use an [-NP-NP] structure to complete the sentence in both prime fragments. The priming effect with the lexically overlapping prime fragment (*showed—showed*) was much larger than that with the non-overlapping prime (*gave—showed*). Similar effects were found in numerous studies (e.g., [14, 18–21]). In a recent meta-analysis of structural priming effects in sentence production, the priming effect with lexical overlap was twice as large as that without lexical overlap [10], which underlined the crucial interaction between lexical access and syntactic encoding in structural priming.

It has been several decades since the debate began about how such a structural priming effect comes about. Pickering and Branigan [12] accounted for structural priming and the lexical boost in terms of a lexicalist model of production [22]. In this model, syntactic information is encompassed in the lemma stratum, wherein a lemma node is linked to combinatorial nodes that represent the syntactic structures licensed by the word. When a speaker comprehends or produces a sentence, the lemma node and the combinatorial node specific to the context are activated and the connection between these nodes is strengthened. The residual activation of the combinatorial node transiently enhances the preference of the prime structure in subsequent processing, which results in structural priming. If the head of the prime sentence is activated again, more activation of the lemma node streams to the combinatorial node. This reinforced activation at the combinatorial node further boosts the preference of the primed structure, causing a lexical boost effect. This residual activation model of structural priming predicts that the effects of structural priming and the lexical boost should decay rapidly. In the lexicalist model [22] that formed the basis of the residual activation model, the activation decays automatically as a power-law function of time. There is no principled reason for the residual activation model to predict that the decay of the activation is modulated by the assignment of attention.

On the other hand, Chang and colleagues [13] posited an implicit learning model of structural priming. The account assumes that speakers make predictions about upcoming utterances. They adjust their syntactic preferences by tuning the weight of the form-meaning mapping each time a prediction error occurs. This adaptation of syntactic preference will be incorporated into the statistical distribution of the form-meaning mapping, which consolidates over time as implicit syntactic knowledge. The implicit learning process in structural priming persists over multiple filler trials (i.e., long-term structural priming; [23–25]). However, the initial implicit learning model of structural priming could not predict the lexical boost effect on structural priming. Chang and colleagues [13] thus tentatively assumed that there is also a mechanism related to explicit memory that is orthogonal to the implicit learning of abstract syntax. This view was later developed into a multifactorial account of structural priming [13, 14, 26].

In such a multifactorial account, apart from the implicit learning processes that essentially underlie the lexical-independent priming, speakers also temporarily store the surface structure and the wording of the prime sentences in explicit memory, possibly for the sake of maintaining/monitoring the coherence in the conversation (see [27] for discussion). The encoded prime sentence can be retrieved in the subsequent production tasks so as to facilitate the syntactic processing, thus contributing to the general structural priming effect. When lexical items are repeated between prime and target, speakers would take the repeated item as a retrieval cue that tracks and reconstructs the information from the prime sentence, which will further

enhance the retrieval of prime structure from explicit memory. Thus, the lexical boost effect is mainly modeled as a lexical cueing effect of explicit memory retrieval. Consistent with the non-binary view of automaticity in language production [8], the multifactorial account predicts that the persistence of syntactic structure involves a tacit, incidental, and automatic procedure (i.e., implicit learning) and an effortful, non-automatic procedure (i.e., an explicit memory-related process).

In Chang and colleagues' initial model, the explicit memory process is mainly proposed as an alternative mechanism that compensates for the insensitivity of the implicit learning model to lexical overlap. Thus, the validity of this postulated explicit memory mechanism in structural priming was initially called into question (e.g., [28]). Nevertheless, a number of studies provide evidence for the possible involvement of explicit memory in structural priming ([14, 29–36]). Importantly, several studies demonstrated that some effects of structural priming show characteristics that are analogous to a short-term memory effect ([14, 31, 36]). Hartsuiker et al. [14] examined the effect of structural priming and the lexical boost in written and spoken dialogue. In two experiments, the number of filler trials between prime and target (i.e., lag) was manipulated. They found that structural priming remained robust for up to 6 intervening trials. However, the effect of lexical overlap, which was significant at Lag 0, quickly diminished when prime and target were separated by two or more filler trials. The rapid decay of the lexical boost effect is analogous to the short-lived memory of sentence structure (e.g., [37]). Branigan and McLean [36] replicated these findings in three- to four-year-old speakers.

Other studies tried to answer a similar question by investigating the effect of structural priming and the lexical boost in people with aphasia ([33, 38, 39]). The basic assumption of these studies is that if lexical overlap functions as a retrieval cue, patients with impaired verbal short-term memory might show more difficulty in maintaining the explicit memory of sentence structure, which should lead to a smaller lexical boost effect in a structural priming task. Man and colleagues' study provided evidence for this assumption: people with aphasia had preserved lexical independent priming but not lexical boost, whether the prime and target were separated by zero or two fillers ([33], but see [38]). These findings suggest that at least a lexical-specific effect of structural priming is driven by a mechanism that rapidly decays, which is constrained by the limited capacity of short-term memory.

Note that it is possible that the retrieval process is not solely dependent on the repeated lexical items. Reasonably, speakers are able to recognize similarities other than lexical repetition (e.g., the events, the configuration of the pictures, and the event structure) between the prime and the target. They might as well employ such repeated representations as a retrieval cue for prime sentences. If this is the case, the decay of explicit memory might also manifest itself in lexical-independent structural priming. This hypothesis has been supported by Bernolet and colleagues [29]. In three pairs of experiments, the researchers investigated the lexical-independent structure repetition of Dutch transitives, datives, and word order in relative clauses. The structure repetition in the experiment was either spontaneous (priming experiment) or instructed (structure memory experiment). They found that the effect of structural priming quickly dropped at Lag 2 and Lag 6, which was comparable to the memory decay of sentence structures. As no lexical item was repeated between prime and target, this fast decay suggested that explicit memory not only contributes to lexical-dependent but also to lexical-independent structural priming. Bernolet and colleagues' finding therefore extends the view of the so-called cueing effect on structural priming. The lexical boost effect cannot be taken as the only index of the postulated explicit memory process in structural priming. Instead, even for minimally related sentences, as long as certain representations are shared between the prime and the target, speakers could retrieve certain fast decaying representation of the prime sentence from memory to facilitate their syntactic choices.

In sum, previous studies suggested two essential properties of the explicit memory process in structural priming: *lexical cueing*, namely the stored sentence structure can be better retrieved when there is lexical overlap between prime and target, and *short longevity*, namely explicit memory effects rapidly dissipate over longer lags. However, although previous studies used these properties to pinpoint the explicit memory process in structural priming, it has not yet been empirically tested whether these two properties are intrinsically driven by explicit memory. First of all, the explicit memory hypothesis proposed that lexical repetition should act as a lexical cue in structural priming that facilitates the memory retrieval of sentence structure. However, to our knowledge, no study has tested the effect of lexical repetition on recalling prior sentence structures. Therefore, it is not clear whether lexical repetition can indeed function as a retrieval cue. Second, the explicit memory hypothesis assumes that the short-term decay of the priming effects is relative to the limited capacity for speakers to maintain syntactic information in explicit memory (see [13, 14]). Although some evidence was found in studies on structural priming in people with aphasia (e.g., [33, 38, 39]), no evidence has been gathered regarding whether limited memory capacity constrains structural priming in healthy speakers.

In response to these lacunae in the evidence we discussed above, the current study asks two questions: First, is the lexical boost effect driven by the lexical cueing on sentence structure retrieval? Second, is the short-term effect of structural priming a function of the capacity of speakers to maintain memory traces of sentence structure? We will briefly discuss these two questions further in the sections below.

## Lexical cueing in syntactic persistence

There is a long history of studies on how lexical cues function in memory retrieval (e.g., [40–50]). A general finding of these studies is that the presence of a word that is semantically or phonologically related to the to-be-recalled item facilitates the retrieval of the encoded memory traces (e.g., [41, 42, 48, 49]). The lexical cue is particularly effective when it occurs in both encoding and retrieval processes (e.g., [46, 50]). These findings are consistent with models of cued recognition/recall in that the features that are shared between a cue and a test item can be employed to facilitate the selection of output during memory retrieval (e.g., [51]).

However, it is still an open question whether lexical cues facilitate the retrieval of sentence structures. Although some studies showed better sentence recall when a lexical-semantic cue was presented ([45, 47]), these studies did not directly investigate how syntactic choices in sentence recall were influenced by lexical overlap. As a precondition of the explicit memory hypothesis, lexical overlap serves as a lexical cue for speakers to retrieve the syntactic memory traces that are possibly stored in short-term explicit memory. Thus, it is important to examine the role of lexical overlap in a task that directly taps into memory retrieval of sentence structure (e.g., sentence structure recall). If there is a facilitation effect of lexical overlap on sentence structure recall, it supports an important precondition for an account by which such a lexical cueing effect explains the lexical boost effect on structural priming. One purpose of the current study is to test this precondition of the explicit memory hypothesis, namely that lexical overlap facilitates the recall of sentence structures.

We therefore conducted two sentence structure memory experiments and two structural priming experiments. In the sentence structure memory experiments, participants were instructed to memorize the structure of the sentences that a confederate of the experimenters produced (i.e., to-be-recalled sentences) and reuse it in the subsequent production task. We manipulated both the structure of the to-be-recalled sentences and the overlap of the head noun in prime/to-be-recalled and target sentences in all four experiments in order to investigate the effect of lexical overlap on the memory retrieval of sentence structure.

## Limited memory capacity in the decay of structural priming

It is well established that the maintenance of the memory traces stored in short-term memory is constrained by the assignment of attentional resources ([52–55]). Barrouillet and colleagues argued that the maintenance of memory traces is a time-based process that requires attention. Items encoded in short-term explicit memory can be refreshed when attention is directed to them. But when attention is switched away to processing, memory suffers from a time-related decay. As the central bottleneck only allows one process at a time, the sharing of attentional resources is realized by a constant switching between processing and memory maintenance. Crucially, Barrouillet and colleagues' model acknowledges the variability of memory decay within a fixed time window: even when the duration of the processing is controlled, a more attention-demanding task might yield a greater detrimental effect on memory maintenance.

If the short-term persistence of syntactic structure is indeed a short-term explicit memory effect, it is reasonable to assume that the decay of this short-term effect is dependent upon the limited resources that can be assigned to the maintenance of the memory of sentence structures ([13, 14]). In this case, the rapid decay of the priming effect is not only a function of the time lag or the number of fillers between prime and target but also relative to the amount of time attention is occupied by memory maintenance. On the basis of Barrouillet and colleagues' model, two more predictions can be made. First, there will be more decay of the priming effect when prime and target are separated by a secondary task that requires more attention, even when the time lag between prime and target is fixed. Second, such modulation of priming by cognitive load occurs regardless of the nature of the secondary task, so the priming effect decays even when no new sentence material is encountered. It is possible that an effect of cognitive overload would affect lexically mediated priming in particular, given that lexical overlap might make the process of sentence structure retrieval from explicit memory more likely. But if, as suggested by [29], the repeated lexical item is not the only trigger of the memory retrieval process in structural priming, we would predict that both lexical-dependent and lexical-independent structural priming would be susceptible to load manipulation.

To test these predictions, we investigated the lexical overlap effect and cognitive load effect on structural priming and sentence structure recall of Dutch genitives (of-genitive vs. s-genitive, see [56, 57]). We employed Dutch genitives as the target structure because the priming of Dutch genitive alternation is a well-established effect with substantial magnitude [10]. Two computer-paced structural priming experiments were conducted, along with two corresponding sentence structure memory experiments ([29, 39]). All experiments had the same design.

Importantly, we inserted a secondary arithmetic problem solving task between prime and target in each trial. We controlled the duration and manipulated the difficulty of the problem solving task. We used an arithmetic problem solving task as the secondary task for mainly three reasons. First, a plethora of studies demonstrated that the processing load of arithmetic problems is a function of the problem difficulty (see [58, 59] for a review). Second, arithmetic problem solving (or the operation solving task) is one of the most frequently used secondary tasks in studies of working memory (e.g., [52, 55]). Third, it has been established that the difficulty of a concurrent arithmetic problem influences the preparation time of sentence production [60]. It is possible that the cognitive load imposed by a difficult arithmetic problem would similarly influence the syntactic choices in sentence production.

In sum, based on the rationale above, we tested two predictions. First, lexical overlap enhances the magnitude of both structural priming and sentence structure recall. Second, structural priming and sentence recall are reduced by a secondary task with high vs. low cognitive load.

## Experiment 1a: Structural priming

### Method

**Participants.** Forty Ghent University students, all native Dutch speakers, participated in exchange for course credit (33 females and 7 males, average age 19.35 years). All participants reported to be non-color-blind and right-handed and had normal or corrected-to-normal vision. A 22-year-old female native Dutch speaker acted as confederate. The study is in line with the "General Ethical Protocol for Scientific Research at the Faculty of Psychology and Educational Sciences of Ghent University" and was approved by the Ethical Committee of Faculty of Psychology and Educational Sciences, Ghent University. Informed consent was obtained for experimentation with human subjects.

**Materials.** A verification set of 120 pictures and a description set of 96 pictures for participants were adopted from Bernolet et al. [29]. All pictures showed black-and-white line drawings of two figures. The participant's description set contained 48 critical description pictures and 48 filler pictures. The critical pictures were designed to elicit genitive expressions (see Fig 1). On each critical picture, the figures were holding an object, wearing an object, or standing next to an object, indicating the status of ownership. One object in the picture was colored; the rest of the picture was in black and white. This way, the referential expression the speakers could choose for the colored object should contain information of the ownership of the object (e.g., *the duck of the boy/the boy's duck is red*). The filler pictures contained no objects but two figures, with one fully colored and the other in black and white. All the figures in the participant's description set were chosen equally often from a boy, a girl, a nurse, a wizard, a pirate, a nun, a priest, and a witch. Four colors (blue, green, red, yellow) were used equally often for the different objects and figures in the pictures. The participant's verification set contained 72 critical pictures and 48 filler pictures. Among the critical pictures in the verification set, 48 pictures matched with the confederate's description (24 for the descriptions that shared the object with the corresponding target picture; 24 for the descriptions containing the object that differed from the corresponding target picture) and 24 pictures differed from the description. The configuration of the pictures was the same as the description set.

A description set of 240 sentences for the confederate was created. Half of the sentences matched with the participants' pictures from the verification set. The confederate's description set contained 192 critical prime sentences that corresponded to 48 critical description pictures for participants and were counterbalanced between prime conditions and head noun

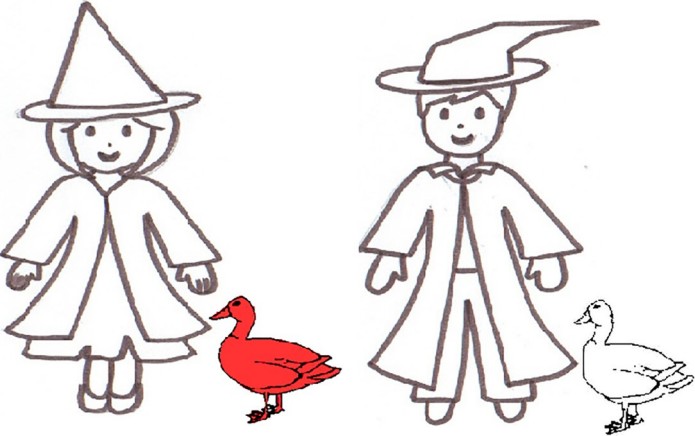

**Fig 1. Example of a target picture.**

conditions. Half of these critical sentences were s-genitive sentences (e.g., 1a and 1b), the other half were of-genitive sentences (e.g., 2a and 2b). The critical prime sentence in the Same Head Noun conditions (e.g., 1a and 2a) contained a head noun (e.g., *eend*, meaning *duck*) that matched with the target object of the corresponding target picture (e.g., a red duck). In the Different Head Noun conditions, the semantic and phonological overlap between the head noun of the prime sentence (e.g., *kaas* in 1b and 2b, meaning *cheese*) and the target object (e.g. a red duck) was kept to a minimum. The prime nouns and their non-overlap controls had the same number of syllables and were matched for prosody. The target objects in the prime and target descriptions had the same color; the owner of the object was different in prime and target descriptions. The remaining 48 sentences in the confederate's description set were filler sentences that could be used to describe the filler items in the participant's verification set. The confederate's verification set contained 96 further pictures that were the same as the pictures in the participant's description set. All materials are listed in S2 Appendix.

 (1a) De jongen zijn eend is rood.
 [Literally: The boy his duck is red.]
 (1b) De jongen zijn kaas is rood.
 [Literally: The boy his cheese is red.]
 (2a) De eend van de jongen is rood.
 [The duck of the boy is red.]
 (2b) De kaas van de jongen is rood.
 [The cheese of the boy is red.]
 *S-genitive, Same Head Noun*
 *S-genitive, Different Head Noun*
 *Of-genitive, Same Head Noun*
 *Of-genitive, Different Head Noun*

 Furthermore, 144 addition problems (S3 Appendix) were constructed for the participant's arithmetic problem solving task. Each problem was composed of two addends and a plus sign. The problem set contained 96 critical problems and 48 filler problems. The problems in the Easy Problem condition were composed of a two-digit addend and a one-digit addend that is either 1 or 2 (e.g., 35 + 2). The problems in the Difficult Problem condition were composed of two two-digit addends (e.g., 35 + 42). In the critical trials, the addition never involved a carrying operation. The units digits of all two-digit addends ranged from 1 to 7, and the tens digits of the smaller two-digit addends ranged from 1 to 4. The order of the addends in each problem was counterbalanced. The filler problems contained 16 addition problems between a two-digit figure and 1 or 2, 16 addition problems between two two-digit figures without carrying, and 16 addition problems between two two-digit figures with carrying. The design of the problems for the confederate was the same as the participant, but the problems in each trial were different between the two.

 A critical trial was defined as a pairing of the confederate's critical prime sentences and the participant's critical description pictures, which were separated by the arithmetic problem solving task. Thus, we had a 2 (prime condition) x 2 (head noun condition) x 2 (problem difficulty) design; all factors were manipulated within items and participants. We constructed eight counterbalanced pseudo-random lists so that each target object was preceded by the same object in four lists (Same Head Noun conditions) and by an unrelated control object (Different Head Noun conditions) in the other four lists. In the Same Head Noun condition and Different Head Noun condition the target picture was preceded by an s-genitive in four lists and by an of-genitive in the other four lists. In four lists, the prime and the target were separated by an easy problem, and in the other four lists, the prime and the target were separated by a difficult problem. For each of the eight lists, the trials were presented in the same pseudo-

random order. There were four fillers at the beginning of each list; critical trials were separated by 0 to 6 filler trials. Each participant was presented with one of these eight lists.

**Procedure.** Both the participant and the confederate sat in front of a PC, which ran the experimental program on Eprime (version 2.0.10.356). They could not see each other's screens. They were told that they would cooperate with their partners to perform as fast and accurately as possible in a series of tasks. In the picture description tasks, one would describe pictures and the other would judge whether the picture on his/her screen matched with the description made by the partner. In the arithmetic problem solving task, one would solve an arithmetic problem and the other would judge the correctness of the result. At the beginning of the experiment, the participants and the confederate read the instruction to learn the series of tasks they would perform and the possible pictures they would see during the experiment. They then familiarized themselves with the procedure in a practice session. The practice session included five filler trials.

The procedure of the main test in Experiment 1a and 1b is illustrated in Fig 2. Note that the timing of each task was strictly controlled. The program was synchronized between the

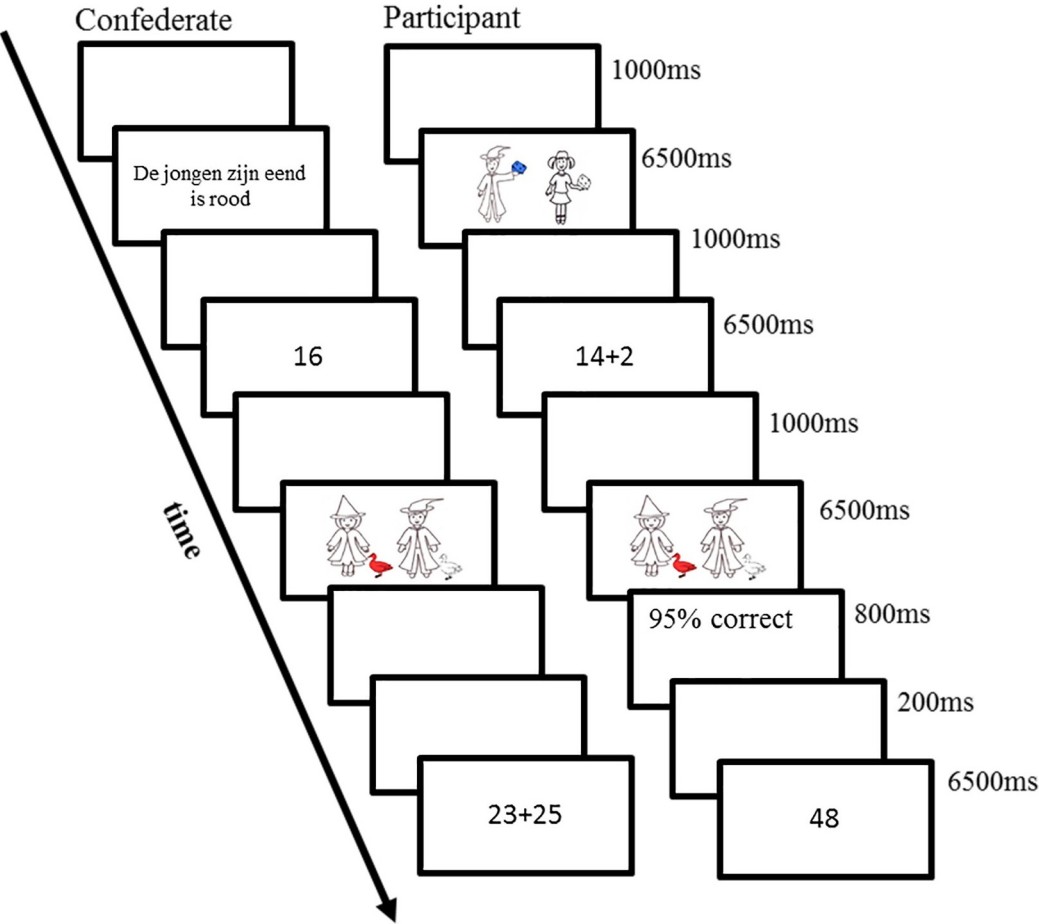

**Fig 2. The procedure in Experiment 1a.** Each trial for participants consisted of the following events: After a 1000 ms interval the prime sentence was presented auditorily (by the confederate) and simultaneously a verification picture was presented (6500 ms). After another 1000 ms interval an arithmetic problem was presented (6500 ms), followed by a 1000 ms interval, a target picture (6500 ms), and feedback (1000 ms). After another 1000 ms interval the verification number was presented (6500 ms). The series of events in Experiment 1b was nearly identical. The only difference was that a parity judgment task (1000 ms) was presented for the confederate instead of an interval after the confederate's verification picture.

participant and the confederate and it was set up so that the confederate always took the first turn. At the beginning of each trial, the confederate started by "describing the picture" while she actually read sentences directly from the screen. The participant pressed "1" if the description matched with the picture and "2" otherwise. After the key was pressed, the picture remained on the screen for 6500 ms. Then an arithmetic problem appeared on the participant's screen. The participant typed in the answer first and then verbally reported the answer. After hearing the answer, the confederate verified the answer by pressing "1" or "2". This task lasted for 4500 ms and a visual signal appeared on the screen for 2000 ms by the end of the task to remind the participants to verbally report the results. The participant then saw a picture on the screen and described it to the confederate. The confederate judged whether the description matched the picture. The task lasted for 6500 ms. Upon responding, the participant saw feedback about his/her average accuracy in the arithmetic problem solving task. The feedback was presented on the screen for 800 ms. Finally, the confederate solved an arithmetic problem and verbally reported the number to the participant. The participant verified the reported result. The experimental session took about 50 minutes.

**Scoring.** The latency of the first key press for participants in the arithmetic problem solving task was recorded as an index of processing time. Responses for participants during the picture description task were coded as s-genitives, of-genitives, or 'others'. A response was coded as an s-genitive if the possessor preceded the possessed object and the appropriate possessive morpheme (*zijn* for male possessor/*haar* for female possessor) was added between the possessor and the object. A response was coded as an of-genitive if the sentence began with the possessed object, followed by the preposition *van*, and ended with the possessor. If a different preposition was used, or if any other construction was used, the response counted as 'other' response.

## Results

Data and analysis scripts for all experiments reported in this paper are available online (on the Open Science Framework at https://osf.io/6utkf/). Critical trials in which participants produced no response or an 'other' response were excluded from the analyses (8.6% of the data). The final data set contained 1755 target responses, among which were 274 s-genitive responses (15.6%) and 1481 of-genitive responses (84.4%).

We first report the results of the arithmetic problem solving task. The average time occupied by problem solving in the Easy Problem condition was 1311 ms shorter than the average time of problem solving in the Difficult Problem condition (Easy: 1548 ms vs. Difficult: 2851 ms, Cohen's d = -3.085), indicating an evident distinction of cognitive load between the two difficulty levels of the secondary task.

The descriptive data of the s-genitive production for each prime condition x head noun condition is illustrated in Table 1. The overall s-genitive production was 26.0% after an s-genitive prime and 5.1% after an of-genitive prime, yielding a 20.9% structural priming effect. More s-genitives were produced in the Same Head Noun condition than in the Different Head Noun condition (20.3% vs. 10.6%), while there was no difference in the s-genitive production between the Easy Problem and Difficult problem condition (15.8% vs. 15.0%). The priming effect was 31.2% in the Same Head Noun condition and 10.2% in the Different Head Noun condition, resulting in a 21.1% lexical boost effect on structural priming. The difference in the priming effect between the Easy Problem and Difficult Problem conditions was negligible in both the Same Head Noun condition (1.1%) and Different Head Noun condition (0.6%). The priming effect for each overlap x problem difficulty condition is illustrated in Fig 3.

The participants' responses were fit by a Generalized Linear Mixed Model (GLM model), using the *lme4* package (version 1.1.23) in *R* (version 3.4.0). The model predicted the logit-

**Table 1. Proportions of s-genitive responses out of all the s-genitive and of-genitive responses for each prime condition\*head noun condition combination in each experiment.**

| Experiment | Head noun condition | Prime/ to-be-recalled structure | | |
| --- | --- | --- | --- | --- |
| | | S-genitive | Of-genitive | Structure repetition |
| EXP1a | Same | .360 | .047 | .313 |
| | Different | .157 | .055 | .102 |
| EXP1b | Same | .784 | .040 | .744 |
| | Different | .530 | .056 | .474 |
| EXP2a | Same | .404 | .022 | .382 |
| | Different | .204 | .041 | .163 |
| EXP2b | Same | .865 | .019 | .846 |
| | Different | .641 | .028 | .613 |

"S-genitive" and "Of-genitive" in the second row of the header indicate the levels of prime condition in Experiment 1a and 2a as well as the levels of to-be-recalled structure in Experiment 1b and 2b.

"Structure repetition" in column 5 refers to the structural priming effect in Experiment 1a and 2a as well as the structure memory effect in Experiment 1b and 2b. It indicates the proportion of s-genitive responses in s-genitive condition (column 3) minus that in of-genitive condition (column 4).

transformed likelihood of an s-genitive response. Prime condition, head noun condition, and problem difficulty were included in the model as fixed factors. These predictors were entered into the model in mean-centered form (deviation coding). We also included the critical trial number (normalized) as an independent variable, as a number of studies suggest a cumulative priming effect, so that the likelihood of production of the least frequent structure gradually increases over the course of the experiment toward the statistical distribution of the current language environment (e.g., [61, 62]). For the analysis (and all the analyses thereafter), we employed the maximal random effects structure justified by the design [63]. Specifically, we included in the model the by-subject and by-item random intercepts as well as random slopes for all main effects and interactions in the fixed model. If the maximal model could not

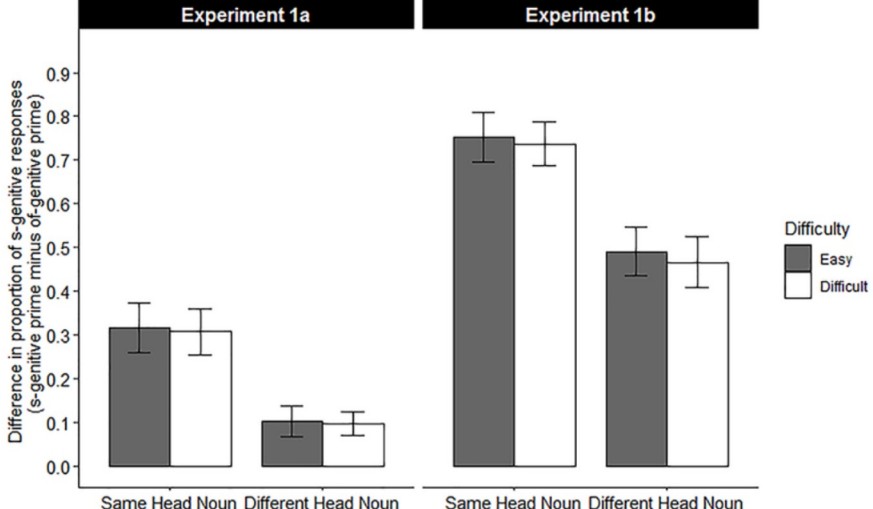

**Fig 3. The priming effect (s-genitive production in the s-genitive condition minus that in the of-genitive condition) as a function of head noun condition and problem difficulty in Experiment 1a and 1b.** Error bars reflect standard errors calculated for a by-participants analysis.

**Table 2. Fixed effect estimates (in log odds units), Experiment 1a.**

|  | Estimate | SE | z | p-value |
|---|---|---|---|---|
| (Intercept) | -3.157 | 0.343 | -9.203 | < .001 |
| Prime condition | 2.918 | 0.288 | 10.121 | < .001 |
| Head noun condition | 0.659 | 0.260 | 2.537 | .011 |
| Problem difficulty | 0.240 | 0.230 | 1.042 | .298 |
| Critical trial number | -0.379 | 0.096 | -3.949 | < .001 |
| Prime condition: Head noun condition | 2.083 | 0.470 | 4.432 | < .001 |
| Prime condition: Problem difficulty | -0.223 | 0.454 | -0.491 | .623 |
| Head noun condition: Problem difficulty | 0.691 | 0.454 | 1.522 | .128 |
| Prime condition: Head noun condition: Problem difficulty | -0.968 | 0.896 | -1.081 | .280 |

Prime condition (Of-genitive as the baseline level), head noun condition (Different Head Noun as the baseline level), problem difficulty (Difficult Problem as the baseline level) were in mean-centered form. Critical trial number was normalized.

converge or showed singularity, we first dropped the random correlation terms once and for all. If the model without random correlation could not converge either, we begin to drop one random factor at a time, starting from the most complex terms in the random model. When there were multiple terms with the same complexity, we compared the variances of the random effects in the last model and dropped the term that accounted for the least amount of variance. We repeated this step until the model converged and no warning of singular fit was reported.

The final model included a random intercept, a random slope of prime condition, and a random slope of head noun condition for subjects as well as a random intercept, a random slope of prime condition, a random slope of head noun condition, and a random slope of problem difficulty for items. The random correlations were dropped. Here we report the significance of the effects based on the fixed effect estimate of the of the LME models. This is because by using contrastive coding, the fixed effects of the model are informative about the main effects and the interactions [64]. The summary of the fixed effects of the model is listed in Table 2. Alpha was assumed as .05. The significant negative intercept ($p_z < .001$) indicates that out of all the s-genitive and of-genitive responses, the proportion of s-genitive was significantly below 50%. To our expectation, we found a significant main effect of prime condition ($p_z < .001$). The significant interaction between prime condition and head noun condition ($p_z < .001$) demonstrated an evident lexical boost effect. Additionally, we found a significant main effect of critical trial number ($p_z < .001$). The negative coefficient indicated that the overall likelihood of s-genitive production decreased over the course of the experiment. However, there was no significant two-way interaction between prime condition and problem difficulty ($p_z = .623$) and neither was there a three-way interaction among prime condition, head noun condition, and problem difficulty ($p_z = .280$). Thus, there was no evidence that secondary task difficulty modulated structural priming or the lexical boost. We also found a significant main effect of head noun condition ($p_z = .011$). Although this effect was not predicted, we argued that this might be a by-product of the lexical boost effect. It is possible that the facilitation effect of the head noun overlap on the persistence of sentence structure further led to an increase of the overall production of the least frequent structure (s-genitive), resulting in a higher likelihood of overall s-genitive production when there was a head overlap. The rest of the effects were not significant ($p_z s > .1$).

There was no effect of cognitive load on structural priming. One possibility is that the arithmetic problems were generally too easy, even in the "difficult" condition, to exert substantial cognitive load. However, considerable variation can be expected among our subjects and

items, so that there may have been a relatively strong load on a subset of the trials. In an exploratory analysis, we therefore considered the reaction times as a proxy for cognitive load in a subset of data. We predicted that in the Difficult Problem condition there would be a negative correlation between reaction time and recall accuracy [53]. We did not make the same prediction in the Easy Problem condition since solving a simple arithmetic problem like "12+1" requires little attention [59]. We fitted a further model that predicted the likelihood of s-genitive production in the subset of data in the Difficult Problem condition, using prime condition, head noun condition, and the processing time (normalized) as predictors. The final model included a random intercept for subjects as well as a random intercept, a random slope of prime condition, and a random slope of head noun condition for items. The random correlations were dropped. The summary of the fixed effects of the model is illustrated in S1 Appendix. The intercept and the main effect of the prime condition were significant ($p_z$s < .001). The interaction between the prime condition and the head noun condition was marginally significant ($p_z$ = .096). Now we found a significant interaction between the prime condition and processing time ($p_z$ = .006). The negative coefficient estimate indicated a negative correlation between the two predictors: the longer the reaction time in the secondary task, the smaller the priming effect afterwards. However, the three-way interaction among prime condition, head noun condition, and the processing time was not significant ($p_z$ = .990). Unexpectedly, we found a significant main effect of the processing time ($p_z$ = .019). The rest of the effects were not significant ($p_z$s > .1).

## Experiment 1b: Sentence structure memory

### Method

**Participants.** Forty further Ghent University students participated in exchange for course credit (34 females and 6 males, average age 19.85 years). All participants were native Dutch speakers, reported to be non-color-blind and right-handed, and had normal or corrected-to-normal vision. The same Dutch speaker as in Experiment 1a was employed as confederate.

**Materials.** The materials were the same as in Experiment 1a. Because the stimuli in Experiment 1b (and also Experiment 2b) did not serve as primes, we use the term 'to-be-recalled sentence' to refer to the sentences provided by the confederate.

**Procedure.** The procedure was similar to that of Experiment 1a, except that the participant and confederate were told that they would each perform an extra memory task in parallel with one of the tasks. At the beginning of the experiment, the experimenter assigned the structure memory task to the participant and the number memory task to the confederate, making it appear as if the tasks were randomly assigned. The participant was then told that in each trial he/she should memorize the sentence structure used by the confederate for picture description and then reuse the same sentence structure to formulate his or her utterances when he/she had to describe the picture in the same trial. The experimenter did not give a further explanation about what "sentence structure" refers to in order to avoid overexposing participants to metalinguistic knowledge about syntax. However, if speakers failed to reuse sentence structure (in this case an s-genitive or an of-genitive) in the practice session, the experimenter gave corrective feedback to the participant until he/she began to reuse sentence structures. Meanwhile, in order to balance the workload between participant and confederate, the confederate was instructed to memorize the parity (i.e., odd or even) of the number reported by the participant and recall the parity in the subsequent task within the same trial. The participant and confederate familiarized themselves with the procedure with five trials in the practice session.

The program was set up so that the confederate always took the first turn. The program ran simultaneously between the participant and the confederate. At the beginning of each trial, the

confederate started by "describing the pictures". The participant pressed the button to judge whether the picture matched with the description, while memorizing the sentence structure the confederate used. Then an arithmetic problem appeared on the participant's screen. The participant typed in the answer first and then verbally reported the answer. After hearing the answer, the confederate made the judgment about the correctness of the answer, while memorizing the parity of the reported number. The participant then saw a picture on the screen and was instructed to describe it by reusing the sentence structure he/she memorized. In the filler trials, there was no structure alternation so the participant was expected to repeat an intransitive structure. The confederate judged the matching of the description with the picture. Then, the confederate recalled the parity of the reported number by pressing "1" for odd numbers and "2" for even numbers. The recall task lasted for 1000 ms. Finally, the confederate solved an arithmetic problem and verbally reported the number to the participant. The participant verified the reported result. The duration and other settings (including the timing of each task) were the same as in Experiment 1a.

**Scoring.**   The scoring was the same as in Experiment 1a.

## Results

Critical trials in which participants produced no response or an 'other' response in the picture description task were excluded (1.4% of the data). The final dataset contained 1893 target responses, among which were 663 s-genitive responses (35.0%) and 1230 of-genitive responses (65.0%).

In Experiment 1b, the average time occupied by problem solving in the Easy Problem condition was 1310 ms shorter than the average time of problem solving in the Difficult Problem condition (Easy: 1643 ms vs. Difficult: 2952 ms, Cohen's d = -3.403), again clearly reflecting a difference in Problem difficulty. The descriptive data are reported in Table 1. The s-genitive production was 65.8% after an s-genitive to-be-recalled sentence and 4.8% after an of-genitive, yielding a 61.0% sentence structure memory effect. Again, the s-genitive production in the Same Head Noun condition was higher than that in the Different Head Noun condition (41.1% vs. 29.2%) whereas there was no difference in s-genitives between the Easy Problem condition and the Difficult Problem condition (35.3% vs. 35.1%). The structure memory effect was 74.5% in the Same Head Noun condition and 47.4% in the Different Head Noun condition, yielding a 27.0% head noun overlap effect on sentence structure memory. The difference between the recall performance in the Easy Problem and the Difficult Problem conditions was 1.4% in the Same Head Noun condition and 2.5% in the Different Head Noun condition. The recall performance in each head noun condition and problem difficulty condition is illustrated in Fig 3.

A GLM model that predicted the likelihood of s-genitive production was fitted. Note that alternatively, we could have taken the number of correct responses as the dependent variable. However, we decided to use the same dependent variable (number of s-genitives) in all GLM models in the current study in order to make the analyses of the priming and memory experiments comparable. The final model included a random intercept, a random slope of to-be-recalled structure, and a random slope of critical trial number for subjects as well as for items. The random correlations were dropped. The fixed effects were reported in Table 3. There was a significant intercept ($p_z < .001$). In line with our expectation, there was a significant main effect of the to-be-recalled structure ($p_z < .001$), indicating that speakers followed the instruction to use the s-genitive structure if the preceding to-be-recalled structure was an s-genitive (i.e., a structure memory effect). There was also a significant interaction between to-be-recalled structure and head noun condition ($p_z < .001$), indicating a head noun effect on recall

**Table 3. Fixed effect estimates (in log odds units), Experiment 1b.**

| | Estimate | SE | z | p-value |
|---|---|---|---|---|
| (Intercept) | -1.692 | 0.279 | -6.073 | < .001 |
| To-be-recalled structure | 5.574 | 0.469 | 11.881 | < .001 |
| Head noun condition | 0.674 | 0.190 | 3.547 | < .001 |
| Problem difficulty | 0.053 | 0.189 | 0.283 | .777 |
| Critical trial number | 0.033 | 0.121 | 0.274 | .784 |
| To-be-recalled structure: Head noun condition | 2.242 | 0.381 | 5.883 | < .001 |
| To-be-recalled structure: Problem difficulty | 0.110 | 0.377 | 0.293 | .770 |
| Head noun condition: Problem difficulty | 0.301 | 0.383 | 0.787 | .431 |
| To-be-recalled structure: Head noun condition: Problem difficulty | -0.510 | 0.755 | -0.676 | .499 |

To-be-recalled structure (Of-genitive as the baseline level), head noun condition (Different Head Noun as the baseline level), problem difficulty (Difficult Problem as the baseline level) were in mean-centered form. Critical trial number was in normalized form.

of s-genitive structure. However, there was no significant two-way interaction between to-be-recalled structure and problem difficulty ($p_z$ = .770), and neither was there a three-way interaction among to-be-recalled structure, head noun condition, and problem difficulty ($p_z$ = .499). Different from Experiment 1a, there was no main effect of the critical trial number ($p_z$ = .784). Similar to Experiment 1a, there was also a main effect of the head noun condition ($p_z$ < .001), which might also be due to the head noun overlap effect on structure memory led to an overall increase of the least frequent structure (s-genitive) when there was a head noun overlap. The rest of the effects were not significant ($p_z$s > .1).

As in Experiment 1a, we ran an exploratory analysis that considered item difficulty. Again, a GLM model was fitted that predicted the likelihood of s-genitive production after processing a difficult secondary task. The final model included a random intercept, a random slope of to-be-recalled structure, and a random slope of head noun condition for subjects as well as a random intercept and a random slope of to-be-recalled structure for items. The summary of the fixed effects of the model is illustrated in S1 Appendix. There were a significant intercept and a significant main effect of the to-be-recalled structure ($p_z$s < .001). This time there was no two-way interaction between to-be-recalled structure and processing time ($p_z$ = .852). And the three-way interaction among prime condition, head noun condition, and the processing time was not significant either ($p_z$ = .112). There was also a main effect of the head noun condition ($p_z$ = .015). The rest of the effects were not significant ($p_z$s > .1).

**Cross-experiment analysis of structural priming effects and lexical boost effects in Experiment 1a and 1b.** To further compare the magnitude of structural priming effects, lexical boost effects, and cognitive load effects between Experiment 1a and 1b, we fitted a GLM model that predicts the likelihood of s-genitive production in Experiment 1a-b. Prime condition (this condition indicates the prime condition in Experiment 1a and the to-be-recalled structure in Experiment 1b), head noun condition, problem difficulty, and experiment were included in the model as fixed factors (all in mean-centered form). The final model included a random intercept, a random slope of prime condition, and a random slope of head noun condition for subjects as well as a random intercept, a random slope of prime condition, a random slope of head noun condition, and a random slope of experiment for items. The random correlations were dropped.

The summary of the fixed effects of the model is listed in Table 4. The intercept was significant ($p_z$ < .001). There was a main effect of experiment ($p_z$ < .001), indicating that the overall proportion of s-genitive was much higher in Experiment 1b than in Experiment 1a. The main

**Table 4. Fixed effect estimates (in log odds units), cross experiment analysis (1a and 1b).**

| | Estimate | SE | z | p-value |
|---|---|---|---|---|
| (Intercept) | -2.323 | 0.211 | -10.992 | < .001 |
| Prime condition | 4.144 | 0.279 | 14.862 | < .001 |
| Head noun condition | 0.648 | 0.153 | 4.224 | < .001 |
| Problem difficulty | 0.134 | 0.142 | 0.946 | .344 |
| Experiment | -1.518 | 0.414 | -3.669 | < .001 |
| Prime condition: Head noun condition | 2.101 | 0.288 | 7.295 | < .001 |
| Prime condition: Problem difficulty | -0.041 | 0.283 | -0.146 | .884 |
| Prime condition: Experiment | -2.456 | 0.475 | -5.174 | < .001 |
| Head noun condition: Problem difficulty | 0.558 | 0.283 | 1.974 | .048 |
| Head noun condition: Experiment | -0.039 | 0.289 | -0.135 | .892 |
| Problem difficulty: Experiment | 0.230 | 0.287 | 0.803 | .422 |
| Prime condition: Head noun condition: Problem difficulty | -0.693 | 0.565 | -1.227 | .220 |
| Prime condition: Head noun condition: Experiment | -0.062 | 0.582 | -0.106 | .915 |
| Prime condition: Problem difficulty: Experiment | -0.487 | 0.573 | -0.849 | .396 |
| Head noun condition: Problem difficulty: Experiment | 0.532 | 0.571 | 0.932 | .351 |
| Prime condition: Head noun condition: Problem difficulty: Experiment | -0.730 | 1.137 | -0.642 | .521 |

Prime condition (Of-genitive as the baseline level), head noun condition (Different Head Noun as the baseline level), problem difficulty (Difficult Problem as the baseline level), experiment (Experiment 1b as the baseline level) were in mean-centered form.

effect of prime condition was significant ($p_z < .001$) and the two-way interaction between prime condition and experiment was significant ($p_z < .001$), suggesting that the effect of structure repetition in Experiment 1b was significantly stronger than that in Experiment 1a. There was a significant interaction between prime condition and head noun condition ($p_z < .001$), but the three-way interaction among prime condition, head noun condition, and experiment was not significant ($p_z = .915$). The two-way interaction between prime condition and problem difficulty and the three-way interaction among prime condition, problem difficulty and experiment were not significant ($p_z s > .1$). The three-way interaction among prime condition, head noun condition, and problem difficulty as well as the four-way interaction among all four predictors were not significant ($p_z s > .1$). There was also a main effect of head noun condition ($p_z < .001$). Unexpectedly, there was a significant interaction between head noun condition and problem difficulty ($p_z = .048$). The rest of the effects were not significant ($p_z s > .1$).

In addition, the theoretically interesting interactions that involved the contrast between the two experiments were examined by estimating the inverse of Bayes factor ($BF_{10}$) using Bayesian Information Criteria. This compares the fit of the data under the alternative hypothesis (i.e., a model with experiment as an interaction term) to the null hypothesis (i.e., a model without experiment as an interaction term). Based on the standard interpretation of the inverse of Bayes factors as evidence for the alternative hypotheses [65], $BF_{10}$ that ranges from 1 to 3 can be taken as weak evidence for the alternative hypothesis. The higher a $BF_{10}$, the more evidence in support of the alternative hypothesis (3–20: positive evidence; 20–150: strong evidence; > 150: very strong evidence).

In the Bayesian analysis, we found an estimated $BF_{10}$ for the two-way interaction between prime condition and experiment that suggested the data were 1017 times more likely to occur under a model including the two-way interaction than under a model without it. There was thus very strong evidence for the alternative hypothesis, namely that there was a difference in the effects of structure repetition between the two experiments. An estimated $BF_{10}$ for the

three-way interaction among prime condition, head noun condition, and experiment indicated that the data were only 0.017 times more likely to occur under a model including the three-way interaction than under a model without it. This suggested strong evidence against the alternative hypothesis that the magnitude of the head noun overlap effect on structure repetition was different between the two experiments. Similarly, strong evidence against the alternative hypothesis was also found for the three-way interaction among prime condition, problem difficulty, and experiment ($BF_{10}$ = 0.023) and for the four-way interaction among prime condition, head noun condition, problem difficulty, and experiment ($BF_{10}$ = 0.020). Taken together, the cross-experiment analysis showed that while the overall proportion of the s-genitive responses and the tendency to repeat a previously experienced structure were significantly different between Experiment 1a and 1b, the magnitude of the head noun overlap effect (and other interactions under test) was not different between the two experiments.

## Discussion of Experiment 1a and Experiment 1b

In Experiment 1a, we found a structural priming effect in dialogue. Specifically, the likelihood for a speaker to spontaneously produce an s-genitive sentence was higher after an s-genitive prime than after an of-genitive prime (21.2% priming effect). Structural priming survived over a non-linguistic task that mostly taps into cognitive processes that are independent of language processing. These findings, in combination with the previous finding of long-term structural priming effect over linguistic intervening tasks (e.g., [14, 23, 24, 66]), suggest that the structural priming effect may persist over at least one intervening task. Additionally, we found a significant effect of the to-be-recalled structure (60.9%) in Experiment 1b, demonstrating that the participants typically reused the sentence structure as instructed. However, the participants did not perform perfectly in the memory experiment: the overall accuracy of sentence structure memory was 80.5%, indicating that misrecall occurred regularly. This is consistent with the findings of Bernolet et al. [29] in which the accuracy of immediate structure recall ranged from 73.3% to 89.6%.

One possible locus of the syntactic persistence in both experiments is that speakers memorized the gender possessive pronouns in the s-genitives (e.g., *zijn*, *haar*) and used these function words as a pointer to guide the sentence structure formulation in the production task. If so, we would expect that when the gender of the possessive pronouns was consistent between prime and target sentences, the structural priming (and memory) effect would be larger. However, adding the interaction between prime structure and gender consistency did not improve the fit of the GLM model for the full data set in Experiments 1a ($\chi$2 = 0.16, df = 1, p = .900) and 1b ($\chi$2 = 0.416, df = 1, p = .519). Thus, it is unlikely that the priming effect in Experiment 1a and memory effect in Experiment 1b were driven by a sentence formulation process that was guided by the retrieval of the primed possessive pronouns.

As expected, there was an effect of head noun overlap on structural priming (21.1% lexical boost effect). This lexical boost effect with Dutch genitives was in line with the numerous studies that observed lexical overlap as a modulating factor on structural priming (e.g., [12, 18]). More importantly, we found a similar effect of lexical overlap on sentence structure recall (27.0% lexical overlap effect). Despite the evident difference of magnitude between the effect of structural priming and the effect of sentence structure memory, the difference caused by lexical overlap was very similar in both experiments. This suggests that the extent to which lexical overlap facilitates spontaneous syntactic repetition was comparable to the extent to which lexical overlap enhances the retrieval of sentence structure from explicit memory.

One unexpected result in the priming experiment was a negative correlation between the critical trial number and the likelihood of s-genitive production. In contrast with the

prediction of the implicit learning account of structural priming that the likelihood of the least frequent structure would increase over the course of the experiment, the number of s-genitive sentences *decreased* with increasing trial number. We will briefly discuss this cumulative effect in the General Discussion.

In both experiments, the average time needed to process a difficult problem was significantly longer than the time needed for an easy problem. This indicated that the difficult problem exerted substantial cognitive load on the participants. We did not find direct evidence for the effect of secondary task difficulty on structural priming. Nevertheless, the negative correlation between the structural priming effect and the time occupied processing the arithmetic task in the Difficult Problem condition is suggestive that the magnitude of priming can be affected by the cognitive load experienced between prime and target processing. Furthermore, the average chance of successful s-genitive structure repetition was close to the chance level in the Different Head Noun condition (52.6%), which, for a recall task, was unexpectedly low. This led us to further suspect that the memory traces of sentence structure can easily dissipate when speakers process the secondary task, regardless of the difficulty of the task (i.e., an across-the-board load effect). Given these results, we predicted that the secondary task imposed a cognitive load on maintaining the memory traces of sentence structure, thus reducing the priming effect and hindering sentence recall and that a considerably more difficult secondary task might lead to a further reduction of priming and recall. We thus designed two further experiments (Experiment 2a and 2b) with a more difficult secondary task, namely addition problems that involved carry operations. Another change, made for practical reasons, was that instead of using an on-site confederate that performed face-to-face interaction with the participants, we took the recording of the confederate in Experiment 1a and 1b as the prime stimuli in the picture verification task for participants.

## Experiment 2a: Structural priming

### Method

**Participants.**    Forty-eight further Ghent University students participated in exchange for course credit (38 females and 10 males, average age 18.56 years). All participants were native Dutch speakers, reported to be non-color-blind and right-handed, and had normal or corrected-to-normal vision.

**Materials.**    The materials were similar to Experiment 1a. Two major changes were made. First, the difficulty of the arithmetic problems in the difficult condition was increased. Ninety-six addition problems were constructed for the critical set of participants' arithmetic problem solving tasks. The problems were similar to the ones of Experiment 1a, but now the addition in the Difficult problems always involved carrying. The unit digits of all the two-digit addends in the Difficult problems ranged from 3 to 8.

Second, instead of scripted sentences read on-site by the confederate, we used 240 audio clips recorded during Experiment 1a as prime stimuli. This makes the priming manipulation very similar to Experiment 1a (same sentences, spoken by the same confederate in a near-identical experimental context) but without the need to involve the (further) confederate. Each clip contained one prime sentence uttered by the confederate. The prime sentence set contained 192 critical prime sentences and 48 filler sentences. The duration of each audio clip was 2000 ms. The intensity of the clips was normalized to 75 dB. The pictures and prime sentences were the same as those of Experiment 1a.

**Procedure.**    The procedure was very similar to that of Experiment 1a, but with a few minor changes. First, participants now used headphones to listen to prime sentences. Second, as we did not employ an on-site interlocutor, we no longer included tasks for the confederate

(solving a mathematical problem, judging the correctness of the participants' problem solving, and judging the matching of the pictures). Third, as the difficulty of the secondary task in the Difficult Problem condition was increased, we prolonged the duration of the arithmetic problem solving task from 4500 ms to 5000 ms.

The participants were told that they would perform a series of tasks. In the picture verification tasks, they would hear the description of pictures from a previous participant, and they should judge whether the description matches the picture on the screen. In the arithmetic problem solving task, they should solve the mathematical problem and type in the answer as fast and accurately as possible. At the beginning of the experiment, the participants adjusted the headphone to a position that was convenient for them but not detached from their ears. Next, the participants had five trials to practice.

At the beginning of each trial, the participant heard an utterance that described a picture via headphones and made a judgment. The picture judgment task lasted for 4000 ms. Then an arithmetic problem appeared on the participant's screen. The participant typed in the answer. The task lasted for 5000 ms. Finally, the description task lasted for 6500 ms. The experiment took about 30 minutes.

**Scoring.**   The scoring was the same as in Experiment 1a.

## Results

Critical trials in which participants produced no response or an 'other' response in the picture description task were excluded (2.2% of the data). The final dataset contained 2252 target responses, among which were 370 s-genitive responses (16.4%) and 1882 of-genitive responses (83.6%).

In Experiment 2a, the average time occupied by problem solving in the Easy Problem condition was 1937 ms shorter than the average problem solving time in the Difficult Problem condition (Easy: 1444 ms vs. Difficult: 3380 ms, Cohen's d = -4.983), indicating an evident difference of cognitive load between levels that is descriptively much larger than in Experiment 1a. The descriptive data are illustrated in Table 1. The s-genitive production was 30.6% after an s-genitive to-be-recalled sentence and 3.2% after an of-genitive, yielding a 27.5% structure repetition effect. The s-genitive production in the Same Head Noun condition was higher than that in the Different Head Noun condition (21.2% vs. 12.2%) whereas there was no difference of s-genitives between the Easy Problem condition and the Difficult Problem condition (16.5% vs. 16.9%). The structure priming effect was 38.2% in the Same Head Noun condition and 16.3% in the Different Head Noun condition, thus there was a 21.9% head noun overlap effect on structural priming. Descriptively, there was 3.8% more priming in the Easy problem than in the Difficult problem conditions. This difference amounted to 3.2% in the Same Head Noun conditions and 4.7% in the Different Head Noun conditions (Fig 4).

We fitted a GLM model that predicts the likelihood of s-genitive production. The model was fitted in the same way as in Experiment 1a. The final model included a random intercept and a random slope of prime condition, a random slope of head noun condition, and a random slope of critical trial number for subjects as well as a random intercept, a random slope of prime condition, and a random slope of head noun condition for items. The random correlations were dropped. The summary of the fixed effects of the model is listed in Table 5. There were a significant intercept, a significant main effect of prime condition, and a significant interaction between prime condition and head noun condition ($p_z$s < .001). Importantly, there was a significant two-way interaction between prime condition and problem difficulty ($p_z$ = .006) as well as a marginal three-way interaction among prime condition, head noun condition, and problem difficulty ($p_z$ = .071). In addition, there was a main effect of critical

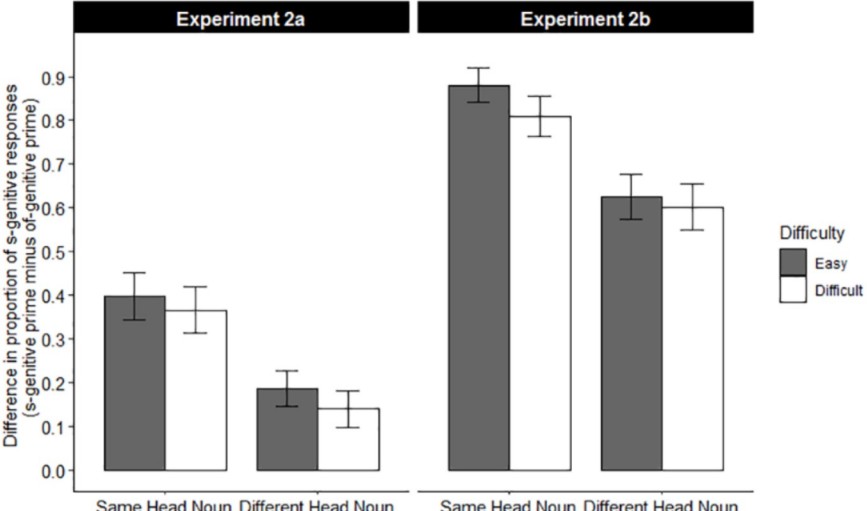

**Fig 4. The priming effect (s-genitive production in s-genitive condition minus that in of-genitive condition) as a function of head noun condition and problem difficulty in Experiment 2a and 2b.** Error bars reflect standard errors calculated for a by-participants analysis.

trial number ($p_z < .001$). The negative coefficient of the fixed effect indicated a decrease of the s-genitive production toward the end of the experiment. Unexpectedly, we also found a significant main effect of problem difficulty ($p_z = .035$) and a significant two-way interaction between problem difficulty and head noun condition ($p_z = .012$). There was no clear theoretical reason for these effects, and similar effects were not found in any other experiments. We decided to refrain from speculation about these unexpected findings, which might well be a result of type I error. The rest of the effects were not significant ($p_z > .1$).

## Experiment 2b: Sentence structure memory

### Method

**Participants.**   Forty-eight further Ghent University students participated in exchange for course credit (39 females and 9 males, average 19.06 years). All participants were native Dutch

**Table 5. Fixed effect estimates (in log odds units), Experiment 2a.**

|  | Estimate | SE | z | p-value |
|---|---|---|---|---|
| (Intercept) | -3.632 | 0.346 | -10.506 | < .001 |
| Prime condition | 4.108 | 0.423 | 9.713 | < .001 |
| Head noun condition | 0.196 | 0.309 | 0.633 | .527 |
| Problem difficulty | -0.611 | 0.289 | -2.113 | .035 |
| Critical trial number | -0.610 | 0.154 | -3.951 | < .001 |
| Prime condition: Head noun condition | 3.001 | 0.588 | 5.102 | < .001 |
| Prime condition: Problem difficulty | 1.571 | 0.577 | 2.725 | .006 |
| Head noun condition: Problem difficulty | -1.452 | 0.578 | -2.512 | .012 |
| Prime condition: Head noun condition: Problem difficulty | 2.081 | 1.152 | 1.807 | .071 |

Prime condition (Of-genitive as the baseline level), head noun condition (Different Head Noun as the baseline level), problem difficulty (Difficult Problem as the baseline level) were in mean-centered form. Critical trial number was normalized.

speakers, reported to be non-color-blind and right-handed, and had normal or corrected-to-normal vision.

**Materials.** The materials were the same as Experiment 1a.

**Procedure.** The procedure was similar to that of Experiment 2a. The only change was that the participants were told that they should memorize the sentence structure used in the audio clip in order to reuse the same structure in the subsequent picture description task. In contrast to Experiment 1b, no parity recall task was presented as this task was only for the confederate in Experiment 1b.

**Scoring.** The scoring was the same as in Experiment 1a.

## Results

Critical trials in which participants produced no response or an 'other' response in the picture description task were excluded (0.7% of the data). The final dataset contained 2287 target responses, among which were 823 s-genitive responses (38.9%) and 1287 of-genitive responses (61.1%).

In Experiment 2b, the average time occupied by problem solving in the Easy Problem condition was 1923 ms shorter than the average problem solving time in the Difficult Problem condition (Easy: 1427 ms vs. Difficult: 3350 ms, Cohen's d = -5.690). Again, there was an evident difference between cognitive load conditions, which was descriptively much larger than in Experiment 1b. The descriptive data of the proportion of s-genitive responses are reported in Table 1. The s-genitive production was 75.4% after an s-genitive to-be-recalled sentence and 2.3% after an of-genitive, yielding a 73.0% sentence structure memory effect. Again, the s-genitive production in the Same Head Noun condition was higher than that in the Different Head Noun condition (44.2% vs. 33.4%) whereas there was no difference in s-genitives between the Easy Problem condition and the Difficult Problem condition (39.4% vs. 38.2%). The structure memory effect was 84.6% in the Same Head Noun condition and 61.4% in the Different Head Noun condition, thus there was a 23.2% head noun overlap effect on sentence structure memory. Descriptively, sentence structure recall was better for easy problems than for difficult problems. This difference amounted to 7.1% in the Same Head Noun conditions and 2.3% in the Different Head Noun condition. (Fig 4).

Again, we fitted a GLM model that predicts the likelihood of s-genitive production. The final model included a random intercept, a random slope of to-be-recalled structure, and a random slope of critical trial number for subjects as well as a random intercept, a random slope of to-be-recalled structure, a random slope of problem difficulty, and a random slope of critical trial number for items. The random correlations were dropped. The summary of the fixed effects of the model is listed in Table 6. There were a significant intercept, a significant main effect of the to-be-recalled structure, and a significant interaction between to-be-recalled structure and head noun condition ($p_z$s < .001). Similar to Experiment 2a, there was also a significant two-way interaction between the to-be-recalled structure and the problem difficulty ($p_z$ = .015). There was a significant three-way interaction among to-be-recalled structure, head noun condition, and problem difficulty ($p_z$ = .037). The main effect of the head noun condition was once again significant ($p_z$ < .001). In addition, we found a main effect of critical trial number ($p_z$ = .048), which had a positive slope. The rest of the effects were not significant ($p_z$s > .1).

To further explore the three-way interaction among prime condition, head noun condition, and problem difficulty, we divided the dataset by the head noun condition and fitted one GLM model for each subset. The GLM model predicted the likelihood of s-genitive production. The to-be-recalled structure and the problem difficulty (all in mean-centered form) were taken as

**Table 6. Fixed effect estimates (in log odds units), Experiment 2b.**

| | Estimate | SE | z | p-value |
|---|---|---|---|---|
| (Intercept) | -1.861 | 0.317 | -5.861 | < .001 |
| To-be-recalled structure | 7.823 | 0.598 | 13.092 | < .001 |
| Head noun condition | 0.925 | 0.238 | 3.884 | < .001 |
| Problem difficulty | -0.017 | 0.233 | -0.074 | .941 |
| Critical trial number | 0.283 | 0.143 | 1.978 | .048 |
| To-be-recalled structure: Head noun condition | 2.853 | 0.475 | 6.007 | < .001 |
| To-be-recalled structure: Problem difficulty | 1.121 | 0.460 | 2.439 | .015 |
| Head noun condition: Problem difficulty | -0.095 | 0.464 | -0.204 | .839 |
| To-be-recalled structure: Head noun condition: Problem difficulty | 1.934 | 0.930 | 2.081 | .037 |

To-be-recalled structure (Of-genitive as the baseline level), head noun condition (Different Head Noun as the baseline level), problem difficulty (Difficult Problem as the baseline level) were in mean-centered form. Critical trial number was in normalized form.

predictors. The summary of the fixed effects of the model is listed in S1 Appendix. The final model for the Same Head Noun subset included a random intercept for subjects. The random correlation was dropped. The intercept was significant ($p_z$ = .001). We found a significant main effect of the to-be-recalled structure ($p_z$ < .001). There was a significant two-way interaction between to-be-recalled structure and problem difficulty ($p_z$ = .012). The rest of the effect was not significant ($p_z$ > .1). The final model for the Different Head Noun subset included a random intercept and a random slope of to-be-recalled structure for subjects as well as a random intercept, a random slope of to-be-recalled structure, and a random slope of problem difficulty for items. The intercept was significant ($p_z$ < .001). We found a significant main effect of the to-be-recalled structure ($p_z$ < .001). There was no significant two-way interaction between to-be-recalled structure and problem difficulty ($p_z$ = .712). The rest of the effect was not significant ($p_z$ > .1).

**Cross-experiment analysis of structural priming effects and lexical boost effects in Experiment 2a and 2b.** To further compare the magnitude of structural priming effects and lexical boost effects between Experiments 2a and 2b, we fitted a GLM model that predicts the likelihood of s-genitive production in Experiments 2a-b. The fixed factors were the same as the ones in the cross-experiment analysis for Experiments 1a-b. The final model included a random intercept, a random slope of prime condition, and a random slope of head noun condition for subjects as well as a random intercept, a random slope of prime condition, a random slope of head noun condition, and a random slope of experiment for items.

The summary of the fixed effects of the model is listed in Table 7. The intercept was significant ($p_z$ < .001). Again, there was a main effect of experiment ($p_z$ < .001). The main effect of prime condition was significant and the two-way interaction between prime condition and experiment was significant ($p_z$s < .001). There was again a significant interaction between prime condition and head noun condition ($p_z$ < .001), but the three-way interaction among prime condition, head noun condition, and experiment was not significant ($p_z$ = .925). The interaction between prime condition and problem difficulty across Experiment 2a and 2b was significant ($p_z$ < .001) and three-way interaction among prime condition, problem difficulty, and experiment was not significant ($p_z$ = .491). The three-way interaction among prime condition, head noun condition, and problem difficulty was significant ($p_z$ = .010), but the interaction between all four predictors was not significant. There was also a significant main effect of head noun condition ($p_z$ = .012). Unexpectedly, there was a marginal main effect of problem difficulty ($p_z$ = .077), a significant interaction between head noun condition and problem

**Table 7. Fixed effect estimates (in log odds units), cross experiment analysis (2a and 2b).**

| | Estimate | SE | z | p-value |
|---|---|---|---|---|
| (Intercept) | -2.603 | 0.225 | -11.553 | < .001 |
| Prime condition | 5.643 | 0.348 | 16.205 | < .001 |
| Head noun condition | 0.498 | 0.199 | 2.503 | .012 |
| Problem difficulty | -0.321 | 0.181 | -1.770 | .077 |
| Experiment | -1.640 | 0.445 | -3.686 | < .001 |
| Prime condition: Head noun condition | 2.838 | 0.370 | 7.668 | < .001 |
| Prime condition: Problem difficulty | 1.286 | 0.363 | 3.548 | < .001 |
| Prime condition: Experiment | -3.454 | 0.590 | -5.851 | < .001 |
| Head noun condition: Problem difficulty | -0.792 | 0.362 | -2.188 | .029 |
| Head noun condition: Experiment | -0.581 | 0.375 | -1.550 | .121 |
| Problem difficulty: Experiment | -0.576 | 0.363 | -1.586 | .113 |
| Prime condition: Head noun condition: Problem difficulty | 1.870 | 0.725 | 2.581 | .010 |
| Prime condition: Head noun condition: Experiment | 0.070 | 0.737 | 0.095 | .925 |
| Prime condition: Problem difficulty: Experiment | 0.499 | 0.724 | 0.689 | .491 |
| Head noun condition: Problem difficulty: Experiment | -1.204 | 0.727 | -1.657 | .097 |
| Prime condition: Head noun condition: Problem difficulty: Experiment | 0.247 | 1.446 | 0.171 | .864 |

Prime condition (Of-genitive as the baseline level), head noun condition (Different Head Noun as the baseline level), problem difficulty (Difficult Problem as the baseline level), experiment (Experiment 2b as the baseline level) were in mean-centered form.

difficulty ($p_z$ = .029), and a marginally significant interaction between head noun condition, problem difficulty, and experiment ($p_z$ = .097). The rest of the effects were not significant ($p_z$s > .1).

In the Bayesian analysis, we found an estimated $BF_{10}$ for the two-way interaction between prime condition and experiment that suggested the data were 460 times more likely to occur under a model including the two-way interaction than a model without it. There was thus very strong evidence for the alternative hypothesis that there was difference in the effects of structure repetition between the two experiments. An estimated $BF_{10}$ for the three-way interaction among prime condition, head noun condition, and experiment indicated that the data were only 0.015 times more likely to occur under a model that includes the three-way interaction than a model without it. This suggested strong evidence against the alternative hypothesis that the magnitude of the head noun overlap effects on structure repetition was different between the two experiments. Similarly, strong evidence against the alternative hypothesis was also found for the three-way interaction among prime condition, problem difficulty, and experiment ($BF_{10}$ = 0.018) and for the four-way interaction among prime condition, head noun condition, problem difficulty, and experiment ($BF_{10}$ = 0.015). Once again, in a cross-experiment analysis of Experiment 2a and 2b, we found that whereas the structure memory experiment showed a larger overall proportion of s-genitive responses and a greater tendency of structure repetition than the structural priming experiment, the effect of head overlap on structure repetition as well as other interactions under test were not different between the two experiments.

## Discussion of Experiment 2a and 2b

In Experiments 2a and 2b, we replicated the findings of Experiments 1a and 1b and obtained clearer evidence for a cognitive load effect on structural priming. We used a more difficult secondary task, which, as we expected, considerably enlarged the difference in reaction times between difficulty conditions (roughly 1300 ms in Experiment 1a-b vs. roughly 1900 ms in

Experiment 2a-b). There were significant effects of sentence structure in both the structural priming experiment (27.0%) and the structure memory experiment (73.1%). Once again, we observed comparable effects of lexical overlap in the structural priming experiment despite the evident difference in the effects of structure repetition. The comparable lexical overlap effect between structural priming and sentence structure memory supports the prediction from the multifactorial account that the lexical boost effect is driven by a short-term explicit memory mechanism that can be magnified by lexical cueing.

In Experiment 2a and 2b, we once again found that the processing of a difficult problem was much slower than that of an easy problem, indicating a considerable amount of cognitive resource is taxed in solving carrying problems. Most importantly, we found that the effect of structural priming was reduced when the target trial was preceded by a difficult secondary task. The maintenance of these memory traces was a function of the difficulty of the secondary tasks (which was directly correlated with the time that the speakers' attention was switched away for concurrent processing), and so was the effect of structural priming. As the duration of the problem solving task is controlled, our study is the first to illustrate the cognitive load effect on sentence structure memory in a fixed time window. This indicated that time lag should not be the only modulating factor of memory decay in structural priming. It is also subject to the limited capacity for speakers to maintain the memory traces of sentence structure.

In Experiment 2a there was a marginal significant three-way interaction among prime condition, head noun condition, and problem difficulty. However, the numerical difference of structural priming effects between problem difficulties in the Same Head Noun condition (3.2%) was very similar to that in the Different Head Noun condition (4.7%). The interpretation of this interaction should be taken with much prudence. This three-way interaction was significant in Experiment 2b. The subset analyses showed that the prime condition x problem difficulty interaction was significant in the Same Head Noun condition, but not in the Different Head Noun condition.

The significant effect of critical trial number was replicated in Experiment 2a: the likelihood of s-genitive production decreased over the course of the experiment. In contrast, in the memory experiment (Experiment 2b), the likelihood of s-genitive production increased with the progress of the experiment, possibly indicating a practice effect. The participants in a sentence structure memory experiment performed in the tasks with an additional goal to accurately repeat the previously experienced sentence structure. It is possible that the accuracy of the structure repetition increased as the participants were getting familiarized with the task, resulting in an increase in the production of the least frequent structure (s-genitive) along with the experiment.

Solving an arithmetic problem mainly taps into the components of the working memory system that assign attentional resources as well as the components that store information ([60, 67]). Some authors argued that two-digit arithmetic solving may involve phonological rehearsal (see [68] for a review), especially when an extra carrying operation is involved [69]. It is possible therefore that in the current experiments, arithmetic problem solving exerted a load not only on the assignment of attention but also on the phonological short-term memory of linguistic representations. However, such a phonological account would predict stronger effects of load in our sentence memory experiments (which involved the explicit instruction to memorize sentence structure and might involve phonological rehearsal) than in our priming experiment (which involved no such instruction). Yet the cognitive load effect was comparable across the priming and memory experiments. But even if the phonological account were correct, it would not take away from our hypothesis that structural priming and sentence structure recall are constrained by limited memory capacity.

## General discussion

In two pairs of experiments, we demonstrated the effects of lexical overlap and cognitive load on structural priming and sentence structure retrieval. In Experiments 1a and 2a, we found stronger structural priming when the head noun overlapped between prime and target, which replicated previous findings of a lexical boost effect on structural priming. The same pattern of effects was also found in Experiments 1b and 2b: The correct recall of the target structure was increased when the head noun was repeated from the encoded sentence. Although the effect of sentence structure was clearly much larger in the sentence structure memory experiments than in the structural priming experiments, the effects of head noun overlap on structure repetition in each pair were comparable. The experiments further showed that the priming effects interacted with cognitive load. In the difficult problem condition of Experiment 1a, the priming effect decreased when the processing time between prime and target increased. More importantly, Experiments 2a-2b demonstrated an effect of cognitive load on both structural priming and sentence memory: there was less priming and poorer recall when the addition problems were more difficult. Below we first discuss the lexical effects on priming and memory, followed by the cognitive load effects. We end with a discussion about the mechanisms of structural priming.

### Lexical cueing effect on sentence structure recall and structural priming

In our sentence structure memory experiments, we set out to test one of the preconditions of the explicit memory hypothesis of structural priming, namely that the memory retrieval of sentence structure can be facilitated by lexical overlap. In both memory experiments, we found that speakers indeed correctly recalled the sentence structure more often when the head noun of the target task repeated that from the prime task. The current study is, to our knowledge, the first study to find an effect of lexical overlap on sentence structure recall. Given that in the current study and previous studies (e.g., [29, 39]), the chance of structural repetition in sentence structure memory experiments was substantially higher than that in structural priming, it is reasonable to assume that sentence structure recall is driven first and foremost by explicit memory of the sentence (and not, for instance, by a process by which the participant bases retrieval on whether a structure is primed). Therefore, the facilitation effect of lexical overlap on sentence recall implied that speakers made use of the repeated lexical item to help them retrieve the representations with which it associated. This lexical cueing effect on sentence structure retrieval converges with previous findings of lexical cueing effects on lexical retrieval (e.g., [46, 50]) and retrieval of sentential information (e.g., [45, 47]).

Importantly, these findings provide an important constraint for theoretical accounts of structural priming. They support one precondition of the explicit memory hypothesis of the lexical boost, namely that the lexical overlap facilitates the recall of sentence structures. In line with the assumptions of the multifactorial account of structural priming (e.g., [13, 14]), we argue that this facilitation is most likely a cue-based memory retrieval effect. The retrieval of lexical items interacts with the retrieval of sentence structures in such a way that the retrieval of the items from the encoded sentence reinforces the retrieval of the sentence structure that is strongly associated with these items during sentence encoding.

In line with previous findings on the effect of verb-particle overlap on idiom priming [32] and the cumulative lexical boost effect [35], we speculate that syntactic information is stored in explicit memory in the form of lexicalized chunks. The entwining between lexical and syntactic information might explain the substantial modulation effect of lexical overlap on memory retrieval of syntactic chunks.

The current study further showed that a similar effect of lexical overlap was also found on structural priming (i.e., lexical boost effect). The priming and memory experiments were the

same in all aspects except for the extent to which explicit memory was taxed. Clearly, the memory experiments require the speakers to retrieve the syntactic structure from short-term explicit memory, but in the priming experiments, speakers were not requested to retrieve the structural information of the sentence. Thus, the most plausible explanation for the resemblance between the lexical overlap effects in the two experiments is that structural priming shares a similar cue-dependent memory retrieval process with sentence structure memory retrieval.

## Limited memory capacity in the persistence of sentence structure

The current study supports the assumption that an attention-demanding short-term memory process contributes to the structural priming effect. First, in the sentence structure memory experiment with a more difficult secondary task (Experiment 2b), we found that the chance of successfully recalling a target structure in the head noun condition was reduced when the production task was preceded by a difficult arithmetic problem. This is compatible with our prediction that the memory traces for lexical-syntactic information suffer from a detrimental effect of a secondary task that demands longer processing time in between memory encoding and retrieval. Thus, the finding supports the view that the maintenance of the memory traces of sentence structure is constrained by limited attentional capacity.

Second, the results further showed a similar effect of cognitive load on structural priming: The general priming effect was reduced by a more attention-consuming secondary task (Experiment 2a). In Experiment 1a, although no significant effect of task difficulty was found, we demonstrated that the priming effect was reduced when speakers' processing time of a difficult problem increased. This implied that there might be a negative association between priming and cognitive load. In Experiment 2a, a load effect on structural priming manifested itself since an extra carry operation was involved in the difficult problem solving. To our knowledge, the current study might be the first to show a cognitive load effect on structural priming in healthy speakers. The cognitive load effect on structural priming indicated that similar to the recall of sentence structure, a short-term memory mechanism contributes to the persistent effect of the prime structure. Such a mechanism is contingent upon the limited capacity for speakers to maintain memory traces of sentence structure: the longer attention is switched away for concurrent processing, the more the memory traces of sentence structure suffer from decay.

The results in Experiment 2a suggested the detrimental effects exerted by cognitive load were numerically similar between the two lexical conditions. So we cannot conclude that the cognitive load effect on structural priming is lexical-dependent. It is somewhat surprising that the cognitive load effect on structural priming was not predominantly lexical-specific. Nevertheless, as Bernolet and colleagues [29] suggested, explicit memory process may also occur in lexical-independent priming, it is reasonable that a higher cognitive load yielded larger interfering effects on the maintenance of sentence structures, even when the head nouns are not repeated.

Furthermore, the lexical boost might be driven by the availability of the primed lexical item stored in explicit memory [27]. As the memory of the content of a sentence is more robust than the form [37], it is possible that after an intervening task, the memory of the primed words stays relatively robust, whereas the memory of sentence structure is prone to the load manipulation. Thus, it is conceivable for a difficult secondary task to be detrimental to general structural priming, while the lexical boost effect remains uninfluenced. Whereas in a sentence memory experiment, speakers primarily maintained the memory traces of sentence structure, such that relatively less attentional resources could be assigned to the maintenance of the

lexical traces including the primed head noun. So that in the memory tasks, the lexical-specific memory of the sentence structure might be more susceptible to the load effect.

Taken together, the current findings are consistent with previous studies on the explicit memory processes in structural priming, in that an explicit memory-related process contributes to both lexical-dependent structural priming ([14, 33, 36]) and lexical-independent structural priming [29].

Combining the findings of cognitive load from two priming experiments, we suggest that structural priming is affected by cognitive load only when the load manipulation is strong enough. This hypothesis might relate the current findings to previous studies that failed to find cognitive load effects on structural priming. In particular, Branigan and colleagues [66] found that the effect of lexical-dependent priming did not differ when prime and target were separated by an intervening sentence as compared to a pure time delay, which might imply that the priming effect stayed robust, regardless of the difference in the cognitive load of the intervening task. One possible reason for the null effect of the intervening sentence is that this secondary task was rather easy (i.e., completing an intransitive sentence) and so it might not have put a strong burden on attention. Thus, the time assigned to maintain the memory traces of sentence structure might not be intrinsically different between an intervening task and a time delay.

This constraint of limited memory capacity on structural priming provides a new perspective on the findings of the rapid decay of the short-term syntactic persistence ([14, 20, 29, 36]). Different from previous studies that examined the time course of structural priming and the lexical boost, the current study did not manipulate the number of fillers between prime and target. Instead, we presented only one filler task with a fixed duration and manipulated its difficulty. The cognitive load effect on structural priming found in the current study suggests that the priming effect is not automatically attenuated with the passage of time but is rather affected by the cognitive load exerted by the processing of materials in between prime and target. This is in line with the working memory models that assume memory load within a given time is modulated by the limited capacity [52–54]. Such a load effect might covary with the time lag, which would then lead to the decay of priming effect over time. In most previous studies, the processing load for the filler tasks (or chunks) in between prime and target was conceivably homogenous. It could be assumed that each filler item required a fixed amount of time during which attention was driven away to processing. Thus, priming effect quickly dissipated over the fillers due to the accumulation of the cognitive load.

## The mechanism of structural priming

In a recent study, [36] nicely recounted the putative cognitive processes of structural priming in children, in which they suggested both automatic processes and non-automatic processes underlie structural priming, including implicit learning, explicit memory, and residual activation. The present paper does not focus on the implicit learning component of structural priming, but we note that in both structural priming experiments the production of the least frequent structure (s-genitive) gradually decreased with the progress of the experiment. This finding seems incompatible with accounts that propose an implicit learning process in structural priming (e.g., [62, 70]). These accounts posited that the cumulative recent experience influences the speakers' structure preference in such a way that the likelihood of producing the least frequent structure would gradually increase as more exemplars of such structure have been mentioned. One speculative explanation of the reversed cumulative effect is that the speakers were most surprised to encounter the s-genitive structures at the beginning of the experiment. They therefore showed the highest possibility to produce the s-genitive structure

early on, because of the relatively large prediction error. As the experiment progressed, the participants gradually regressed to the more default expression, perhaps to alleviate the processing load in sentence production, resulting in the decreasing pattern of the least frequent structure. Note that similar patterns were also found in some cross-linguistic structural priming studies (e.g., [57, 71]). A question for further research is whether it is possible to reconcile implicit learning accounts with such reversed cumulative priming effects.

A main goal of our study was to pinpoint the non-automatic components from structural priming. Our finding of a cognitive load effect on structural priming is best explained by a short-term explicit memory mechanism. Evidence for such an effect was found in both a dialogue experiment and a monologue experiment, which indicates that this short-term memory process in structural priming is context-general. These findings, in combination with the observation of a lexical modulation effect on sentence structure retrieval, suggest that both *lexical cueing* and *fast decay* in structural priming can be at least partially explained by a short-term explicit memory effect. We do not necessarily argue against a possible role of residual activation in structural priming: There may certainly be residual activation from the lemma nodes and combinatorial nodes that shortly boosts the chance that speakers spontaneously repeat syntactic choices along with the explicit memory of sentence structure. Nevertheless, we champion a multi-factorial mechanism of structural priming that at least incorporates lexically dependent short-term explicit memory processes as an essential contributor to the general priming effect (e.g., [13, 14, 36]).

## Conclusions

The current study addressed two questions with respect to the explicit memory mechanism of structural priming. First, our experiments found lexical cueing effects on sentence structure recall, with comparable magnitude to that of a lexical boost effect on structural priming. This supports a crucial precondition for the account that the lexical boost effect in structural priming is driven by cue-dependent memory retrieval. Second, when the manipulation of the cognitive load was strong enough, there was a cognitive load effect on structural priming. This pinpoints a resource-consuming memory maintenance process in structural priming. The findings jointly suggest that the lexical boost effect and short-term decay in structural priming entail the involvement of an explicit memory-related process, supporting an account of structural priming that subsumes multiple memory mechanisms. More broadly, our findings are compatible with the view that speakers can use a capacity-constrained explicit memory to temporarily store a sentence's structure and recycle that structure in sentence production.

## Supporting information

**S1 Appendix. Summary of fixed effects in LME models in the subset analyses of Experiment 1a-b and 2a-b.**
(DOCX)

**S2 Appendix. Primes and targets used in each experiment.** The description of the target picture is given in first line depicts the content of the target picture. The possessor of the colored object and the object that is owned are mentioned. In the following lines, the s-genitive (a) and the of-genitive primes (b) are given in Dutch. In each prime sentence, the noun in the Same Head Noun condition is mentioned before the slash and the noun in the Different Head Noun condition is mentioned after the slash.
(DOCX)

**S3 Appendix. Arithmetic problems used in each experiment.** The appendix only displays each problem in an addend order that the first addend is always larger than the second. In the reported experiment the order of the addends was counterbalanced.
(DOCX)

# Acknowledgments

We would like to thank Nolwenn Dierck for her assistance in data collection.

# Author Contributions

**Conceptualization:** Chi Zhang, Sarah Bernolet, Robert J. Hartsuiker.

**Data curation:** Chi Zhang.

**Formal analysis:** Chi Zhang.

**Investigation:** Chi Zhang.

**Methodology:** Chi Zhang.

**Supervision:** Sarah Bernolet, Robert J. Hartsuiker.

**Visualization:** Chi Zhang.

**Writing – original draft:** Chi Zhang.

**Writing – review & editing:** Sarah Bernolet, Robert J. Hartsuiker.

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
