## [Decision Letter · Decision Letter 0]

7 Apr 2020

PONE-D-20-04033

The role of explicit memory in syntactic persistence: effects of lexical cuing and load on sentence memory and sentence production

PLOS ONE

Dear Mr. Zhang,

I am writing with regard to your manuscript, "The role of explicit memory in syntactic persistence: effects of lexical cuing and load on sentence memory and sentence production" (PONE-D-20-04033). I have received comments from two experts in this area of research, and I have also reviewed your manuscript myself. The reviewers and I find that your experiments are tackling a critical and timely question about the mechanisms that underlie structural priming. These results will be of interest to many in this research community, and as such I would like to see them appear in the literature.

The reviewers have raised a number of issues that will need to be addressed in a revision of your manuscript. The most critical of these is raised by Reviewer 1. The reviewer notes that there seem to be inconsistencies between the way that the analyses are described in the manuscript and the analyses that were actually done. Further, the reviewer has a question about the results of Experiment 2a. To add to the reviewer's comments, you might consider generating estimated means from the model for Experiment 2a to see how well they match the observed means. As always, you should also address the reviewers' other comments in some manner -- either through a revision of the manuscript, or through your response letter. 

-------------------

We would appreciate receiving your revised manuscript by May 22 2020 11:59PM. To enhance the reproducibility of your results, we recommend that if applicable you deposit your laboratory protocols in protocols.io, where a protocol can be assigned its own identifier (DOI) such that it can be cited independently in the future. For instructions see: http://journals.plos.org/plosone/s/submission-guidelines#loc-laboratory-protocols

We look forward to receiving your revised manuscript.

Kind regards,

Mike Kaschak

Academic Editor

PLOS ONE

Journal Requirements:

2. Please upload a copy of Supporting Information Figure S1 Fig 1., S2 Fig 2., S3 Fig 3., S4 Fig 4. which you refer to in your text on page 52.

3. Your ethics statement must appear in the Methods section of your manuscript. If your ethics statement is written in any section besides the Methods, please move it to the Methods section and delete it from any other section. Please also ensure that your ethics statement is included in your manuscript, as the ethics section of your online submission will not be published alongside your manuscript.

Reviewers' comments:

Reviewer's Responses to Questions

**Comments to the Author**

1. Is the manuscript technically sound, and do the data support the conclusions?

Reviewer #1: Partly

Reviewer #2: Yes

2. Has the statistical analysis been performed appropriately and rigorously? 

Reviewer #1: No

Reviewer #2: Yes

3. Have the authors made all data underlying the findings in their manuscript fully available?

Reviewer #1: Yes

Reviewer #2: Yes

4. Is the manuscript presented in an intelligible fashion and written in standard English?

Reviewer #1: Yes

Reviewer #2: Yes

5. Review Comments to the Author

Reviewer #1: This paper investigates the contribution of explicit memory to structural priming. The authors describe four experiments in which they investigate the relationship between structural priming, lexical boost, and the difficulty of a concurrent memory load. In two experiments (one dialogue, one monologue), participants performed a standard structural priming task, describing pictures using one of two genitive alternations. In the other two experiments (again one dialogue, one monologue), participants were explicitly told to repeat the sentence structure of the preceding sentence. In all experiments, in between the prime and target sentence, participants solved an addition problem, which was easy or hard; the difference in difficulty between the two types of math problems was increased for the third and fourth experiment.

A robust structural priming and lexical boost effect was found. The authors also found some evidence of task difficulty – i.e., degree of cognitive load – interacting with degree of syntactic priming and lexical boost, to the effect that subjects were less likely to repeat the prime structure when they cognitive load was more difficult, either because they spent longer engaged in the cognitive load task (exploratory RT analyses) or it was a more difficult math problem.

This is an interesting and well-written paper which addresses a theoretically relevant question about the mechanisms underlying structural priming and its reliance on explicit memory. The research questions are well-grounded in prior research and the experiments seem well-designed. The discussion and interpretation is well-reasoned from the results and interesting as well. However, I think there are issues with the statistics which call into question the main result – that higher cognitive load causes reduced structural priming/recall – and thus I do not think much can be inferred from the results and concomitant conclusions of this paper. I believe a complete re-analysis of the statistics is required, and that may (in my opinion, likely will) completely change the central conclusion of the paper.

Major comments:

-Table 1: This table seems to show that in the Same Head Noun condition, 36.0% of the subjects’ productions were s-genitive and 4.7% were of-genitive, and 15.7% and 5.5%, respectively in the Different Head Noun condition.

(a) Why are these values so low? That means that subjects were producing non-s/of-genitive structures on 60-80% of the trials (depending on condition), correct? What were these other structures?

(b) I can’t reconcile these data with the earlier statements that only 8.6% of responses were “other” and that 84.4% of responses were of-genitive (it looks like from Table 1 that ~5% in each condition were of-genitive).

(c) If subjects are producing a different structure on such a large proportion of the trials, please give some assurance that their behavior is indicative of a larger priming processing effect given how many trials were UNaffected by the prime manipulation.

-The authors write (pg. 20, line 473-476): “we included in the model the by-subject and by-item random intercepts as well as random slopes for all main effects and interactions in the fixed model. The full model (and all the other full models in this paper) converged.” But when I look at the code posted on OSF (thank you!), this does not seem to be the case. There are a ton of models there, but it looks like the full model that is being used in the model comparison ANOVAs is DP0122, which has all random slopes removed (i.e., everything except the intercept). It has the full set of fixed effects, but the random effects are just:

(1 | Subject) + (1 | Item)

Similarly, for all of the sub-models that were run, with the fixed effect in question removed, also have the random effects structure of just (1 | Subject) + (1 | Item).

(a) The authors shouldn’t write that all random slopes were included, as it appears that was not the case for the actual model comparisons. It should be clear as to the model structure that was used for the statistics that are reported.

(b) If the models that included all the random slopes converged, then why is that model not being used for the statistics? Removing random slopes can make a model output anti-conservative and may skew the results.

(c) Note that this point is relevant to all of the experiments, as the code that was posted said it was used for all four experiments.

-Likely related to the previous point, I find the results for Experiment 2a to be rather surprising, given the figures and data. I’m not sure what the error bars represent, but assuming they’re standard error, I find it quite unlikely that those would produce an interaction between Prime Condition x Difficulty or a main effect of Difficulty, and yet those are robustly significant.

When I run the code using the model with no random slopes (DP0122), there is indeed a significant Prime Condition x Difficulty interaction. However, I could not get the full model to converge when I ran it myself (the maximal model with all random slopes + intercepts which the text said had converged). So to try to approximate that model, I ran an F1 ANOVA, which, given that there was a balanced design and minimal data loss (2.2%), and I wouldn’t expect big item differences in your stimuli, the ANOVA should show pretty similar results to the LMER. Some results were very similar, as expected (Prime Condition, Prime Condition x Head Noun Condition) but all of the effects involving Task Difficulty were extremely different, e.g. the main effect of Task Difficulty in the ANOVA was F < 1, p = .92; Prime Condition x Task Difficulty was F = 2.5, p = .12; Prime Condition x Head Noun Condition x Task Difficulty was F < 1, p = .75, and these results seem much more in line with the descriptive data. I am not suggesting that the authors run ANOVAs instead of LMERs; there are many good reasons to run LMERs for this type of data. However, it’s important to make sure the results make sense given what the raw data show. As is noted in the next paragraph: “Unexpectedly, we also found a significant main effect of task difficulty (χ2 = 4.966, df = 1, p = .025) and a significant two-way interaction between task difficulty and lexical overlap (χ2 = 7.213, df = 1, p = .007). There was no clear theoretical reason for these effects, and similar effects were not found in any other experiments. We decided to refrain from speculation about these unexpected findings, which might well be a result of type I error.” Given that Task Difficulty shows a tiny effect in the numerical data, but yet shows a significant (or marginal) result for all of the fixed effects that it's involved in, plus none of them were significant in the sample ANOVA that I ran, and you have the minimal random effects structure in your models, and you even note that some of the Task Difficulty effects were unexpected and might be type I error – I think there is something wrong in the LMERs as conducted. It just doesn’t pass the eyeball test. As I cannot get the maximal models to converge on my machine, I can’t see what result that returns myself, so the advice I have is two-fold:

(a) look into the analyses and really make sure they’re being done correctly and that the statistical results match common-sense intuition based on the numerical data

(b) run the model-comparison stats on the maximal model that was reported to have converged, and see if the effects still hold as currently reported.

-pg. 38, line 924-925 says: “The non-significant interaction between cognitive load and lexical overlap on structural priming was somewhat surprising…” and then discusses the implications of this non-effect. I assume that is referring to the interaction between Task Difficulty x Head Noun Condition in Experiment 2a. However, when reporting the Exp 2a results, the text says (line 740): “Unexpectedly, we also found … a significant two-way interaction between task difficulty and lexical overlap (χ2 = 7.213, df = 1, p = .007).” Please resolve this conflict or clarify which comparison you are referring to in the GD.

Minor comments:

-pg. 25, line 593: The exploratory analysis of Exp 1b investigating processing time (I assume also by using RT, as in Exp 1a) x to-be-recalled structure has df = 0.42. I do not understand how doing a model-comparison test can produce a non-integer degrees of freedom. (Unfortunately I can’t look at the model output to help me understand as I can’t find it in the submission.)

-There are a few times throughout the paper when results between experiments are compared numerically, but not statistically (e.g., pg. 26, comparing the size of the lexical boost effect in Exps 1a vs 1b, etc.). Please make those comparisons statistically explicit and report the significance/non-significance of the cross-experiment comparison.

-I wonder how difficult even the most difficult math problems were. Is carrying in addition really such a challenging task? Subjects are only spending 3 seconds doing the math problem in the hardest case. Please talk a little about why you think this might be enough of a cognitive load to disrupt processing.

-In only Exp 2b is there the expected context effect, whereby production of the less-common structure increased over the course of the experiment; in the other experiments, this effect goes in the opposite direction (the less-common structure decreases over the experiment). This is surprising, given that it’s the opposite direction as would be expected. The authors write that the increase effect (the expected direction) is “possibly indicating a practice effect.” Please discuss this further. Why would there be a practice effect in Exp 2b but not the other experiments with (approximately) the same task? Why would the other experiments show the opposite direction as previously-attested context effects?

Textual/editing comments:

-The references are not all standardized – some use the journal abbreviation (J Mem Lang) and others the full name (Journal of experimental psychology: Learning, memory, and cognition).

-pg. 14: references example stimuli sentences before the sentences are produced in the text. This makes that entire paragraph difficult to follow; I suggest moving the sample sentences before the paragraph describing the stimuli conditions (i.e., at line 330).

-pg. 17-18, lines 419-427: There’s a strange paragraph, maybe it’s supposed to be the figure caption for Figure 2? It’s presented in the middle of the text (with no accompanying figure, which is only at the end of the text body).

-pg. 21, line 501 (same for Exp 1b): The authors write that the model for the supplementary analysis investigating RT of Exp 1a is in the Supplementary Material, but it’s not in Appendix A, B, or C (which are the only attached materials), so please include an explicit pointer to where I can find these results.

-Please note in the figure captions what the error bars are: standard error, standard deviation, confidence interval, etc.

-Please make the y-axis for the two figures on the same scale. It makes it easier to compare the results across experiments.

-Grammar/wording comments:

-pg. 5, line 102-3: “It has been several decades for the debate about how such structural priming effect comes about.” -> “It has been several decades since the debate began about…” (for example)

-pg. 7, line 164: “tried to answer A similar question”

-pg. 9, line 224: “These findings are coherent with…” consistent with?

-pg. 13, line 310, 318: “figurines” -> figures

-pg. 27, line 651: “we predicted that the secondary task posited a cognitive load” -> imposed a cognitive load?

-pg. 30, line 726: “minuses” -> minus

-pg. 35, line 856: “more often correctly” -> delete “correctly” perhaps?

-pg. 37, line 881: intertwinement -> entwining

-pg. 38, line 915-956: “similar as for the recall of sentence structure” -> “similar TO” ?

-pg. 39, line 932: “whereas the memory of sentence structure is prune to the load manipulation.” -> I am not sure what this sentence intends.

Reviewer #2: • I appreciate the availability of the authors’ data.

• Overall, I agree with the authors that it is interesting to further explore cognitive load factors that affect priming. I believe the manuscript presents a nice beginning to this. I am, however, not convinced by Experiment 1a in this manuscript. The authors employ a confederate, which I appreciate. I think it is prudent to consider social factors. Perhaps some participants were high self-monitors and simply took more time answering questions to not be judged poorly by their experimental partner. If this were a factor, it may be that they were paying less attention to their partner’s descriptions in anticipation of their turns.

• Regarding the second series of experiments – I found it interesting that the authors decided to make two major changes – 1) the difficulty of the problems and 2) the dismissal of the confederate. I agree that the arithmetic problems, especially the “easy” ones, likely presented minimal, if any cognitive load on the participants. However, making two changes at once leads to questionable conclusions – how can you be sure which manipulation led to the change in significance?

• I think this is a useful addition to the literature. However, I would recommend running a third study with a confederate and the more difficult problems. It would round off these series of experiments nicely and address the concerns I (and the authors even noted) have about the manuscript.

6. PLOS authors have the option to publish the peer review history of their article (what does this mean?). If published, this will include your full peer review and any attached files.

Reviewer #1: No

Reviewer #2: No

---

## [Author Response · Author response to Decision Letter 0]

19 May 2020

Reviewer #1: This paper investigates the contribution of explicit memory to structural priming. The authors describe four experiments in which they investigate the relationship between structural priming, lexical boost, and the difficulty of a concurrent memory load. In two experiments (one dialogue, one monologue), participants performed a standard structural priming task, describing pictures using one of two genitive alternations. In the other two experiments (again one dialogue, one monologue), participants were explicitly told to repeat the sentence structure of the preceding sentence. In all experiments, in between the prime and target sentence, participants solved an addition problem, which was easy or hard; the difference in difficulty between the two types of math problems was increased for the third and fourth experiment.

A robust structural priming and lexical boost effect was found. The authors also found some evidence of task difficulty – i.e., degree of cognitive load – interacting with degree of syntactic priming and lexical boost, to the effect that subjects were less likely to repeat the prime structure when they cognitive load was more difficult, either because they spent longer engaged in the cognitive load task (exploratory RT analyses) or it was a more difficult math problem.

This is an interesting and well-written paper which addresses a theoretically relevant question about the mechanisms underlying structural priming and its reliance on explicit memory. The research questions are well-grounded in prior research and the experiments seem well-designed. The discussion and interpretation is well-reasoned from the results and interesting as well. However, I think there are issues with the statistics which call into question the main result – that higher cognitive load causes reduced structural priming/recall – and thus I do not think much can be inferred from the results and concomitant conclusions of this paper. I believe a complete re-analysis of the statistics is required, and that may (in my opinion, likely will) completely change the central conclusion of the paper.

Major comments:

#1 Table 1: This table seems to show that in the Same Head Noun condition, 36.0% of the subjects’ productions were s-genitive and 4.7% were of-genitive, and 15.7% and 5.5%, respectively in the Different Head Noun condition.

(a) Why are these values so low? That means that subjects were producing non-s/of-genitive structures on 60-80% of the trials (depending on condition), correct? What were these other structures?

(b) I can’t reconcile these data with the earlier statements that only 8.6% of responses were “other” and that 84.4% of responses were of-genitive (it looks like from Table 1 that ~5% in each condition were of-genitive).

(c) If subjects are producing a different structure on such a large proportion of the trials, please give some assurance that their behavior is indicative of a larger priming processing effect given how many trials were UNaffected by the prime manipulation.

>>Author’s response: These comments stem from a misinterpretation of Table 1, presumably due to our unclear title of the table and labeling of the columns. The Table lists the proportion of s-genitives out of all s-genitives and of-genitives, for each experiment, level of lexical overlap condition (rows), and level of prime condition (columns). Thus, the value of 36% the reviewer is referring to means that in the Same Head Noun, s-genitive prime condition, 36% of valid responses were s-genitives (and so 64% were of-genitives). But in the Same Head Noun, of-genitive prime condition, only 4.7% of the responses were s-genitives (and so 95.3% were of-genitives). Thus, there was a priming effect of about 30%. This of course also means that the number of others was really 8.6% - the proportions listed in the Table are based only on non-others (s-genitives and of-genitives). We have revised the title and the header of the table. We hope this clarifies the table.<<

#2 The authors write (pg. 20, line 473-476): “we included in the model the by-subject and by-item random intercepts as well as random slopes for all main effects and interactions in the fixed model. The full model (and all the other full models in this paper) converged.” But when I look at the code posted on OSF (thank you!), this does not seem to be the case. There are a ton of models there, but it looks like the full model that is being used in the model comparison ANOVAs is DP0122, which has all random slopes removed (i.e., everything except the intercept). It has the full set of fixed effects, but the random effects are just:

(1 | Subject) + (1 | Item)

Similarly, for all of the sub-models that were run, with the fixed effect in question removed, also have the random effects structure of just (1 | Subject) + (1 | Item).

(a) The authors shouldn’t write that all random slopes were included, as it appears that was not the case for the actual model comparisons. It should be clear as to the model structure that was used for the statistics that are reported.

(b) If the models that included all the random slopes converged, then why is that model not being used for the statistics? Removing random slopes can make a model output anti-conservative and may skew the results.

(c) Note that this point is relevant to all of the experiments, as the code that was posted said it was used for all four experiments.

>>Author’s response: We thank the reviewer for pointing this out. In fact, the script in the OSF data repository contained an older version of our analysis, in which we employed backward selection for the models. In the version of the analyses that we reported in the manuscript we did employ the maximal random model suggested by Barr et al. (2013). We are very sorry that we did not update the OSF files in time. We have now put the correct R scripts in the OSF repository, one for each pair of experiments. In the script we included all the necessary codes for the descriptive and inferential analyses in the manuscript.<<

#3 Likely related to the previous point, I find the results for Experiment 2a to be rather surprising, given the figures and data. I’m not sure what the error bars represent, but assuming they’re standard error, I find it quite unlikely that those would produce an interaction between Prime Condition x Difficulty or a main effect of Difficulty, and yet those are robustly significant.

>>Author's response: The error bars denote the standard of the error mean from a by-participants analysis; this has now been added to the Figure caption. In interpreting the figure, please keep in mind that the Y-axis represents the priming effect. Thus, the interaction between prime condition and difficulty is reflected in a difference between the easy and difficult conditions. The main effect of difficulty (which is of no theoretical interest) cannot be gauged from the figure.<<

When I run the code using the model with no random slopes (DP0122), there is indeed a significant Prime Condition x Difficulty interaction. However, I could not get the full model to converge when I ran it myself (the maximal model with all random slopes + intercepts which the text said had converged). So to try to approximate that model, I ran an F1 ANOVA, which, given that there was a balanced design and minimal data loss (2.2%), and I wouldn’t expect big item differences in your stimuli, the ANOVA should show pretty similar results to the LMER. Some results were very similar, as expected (Prime Condition, Prime Condition x Head Noun Condition) but all of the effects involving Task Difficulty were extremely different, e.g. the main effect of Task Difficulty in the ANOVA was F < 1, p = .92; Prime Condition x Task Difficulty was F = 2.5, p = .12; Prime Condition x Head Noun Condition x Task Difficulty was F < 1, p = .75, and these results seem much more in line with the descriptive data. I am not suggesting that the authors run ANOVAs instead of LMERs; there are many good reasons to run LMERs for this type of data. However, it’s important to make sure the results make sense given what the raw data show. As is noted in the next paragraph: “Unexpectedly, we also found a significant main effect of task difficulty (χ2 = 4.966, df = 1, p = .025) and a significant two-way interaction between task difficulty and lexical overlap (χ2 = 7.213, df = 1, p = .007). There was no clear theoretical reason for these effects, and similar effects were not found in any other experiments. We decided to refrain from speculation about these unexpected findings, which might well be a result of type I error.” Given that Task Difficulty shows a tiny effect in the numerical data, but yet shows a significant (or marginal) result for all of the fixed effects that it's involved in, plus none of them were significant in the sample ANOVA that I ran, and you have the minimal random effects structure in your models, and you even note that some of the Task Difficulty effects were unexpected and might be type I error – I think there is something wrong in the LMERs as conducted. It just doesn’t pass the eyeball test. As I cannot get the maximal models to converge on my machine, I can’t see what result that returns myself, so the advice I have is two-fold:

(a) look into the analyses and really make sure they’re being done correctly and that the statistical results match common-sense intuition based on the numerical data

(b) run the model-comparison stats on the maximal model that was reported to have converged, and see if the effects still hold as currently reported.

>>Author’s response: We appreciate it that the reviewer went to the trouble of checking the analyses and even conducted an ANOVA as an approximation. We apologize again that the scripts on OSF were outdated. We have checked all analyses, and observed that our statistical inferences regarding task difficulty do hold - although these reanalyses did bring up a new issue, namely that of singular fit, which led to a number of changes to the inferential statistics, and for one interaction a difference in significance (see below). We respond in more detail below:

(a) Model convergence

In the data analyses employed in the manuscript we kept the maximal random model for each linear mixed model. We accepted the model results that showed the warning singular fit since the model converged. But experts on linear mixed models have raised caution about the singular fit in model convergence. Some argued it might indicate that a complicated model is overfitted and might result in mis-convergence (Bates et al., 2015). Thus, in the current version of the data analyses, we followed Barr et al. (2013) by keeping the most complex random model that did not result in singular fit. To do this, we started with the maximal random model and simplified the random model only when the model failed to converge or the model generated a singular fit. If the model showed singularity, we first dropped the random correlations then dropped one random factor a time, starting from the most complex interaction term, until the non-singular model was fitted. We updated the modelling method as well as the structure and fixed effects of the final models in the manuscript and appendices.

(b) The three-way interaction in Experiment 2a

The only place where there might be a discrepancy between the observed data and the model estimation was in the three-way interaction between prime condition, head noun condition, and problem difficulty in Experiment 2a. Descriptively, the difficult problem reduced the numerical size of the structural priming effect to a similar extent in the Same Head Noun condition and in the Different Head Noun condition. However, in the latest data analysis, the three-way interaction became significant in the model comparison analysis (i.e., model comparison between the reduced model and the full model). A possible interpretation of the three-way interaction is that the two-way interaction between prime condition and problem difficulty differed in the subsets divided by head noun condition. Although we did not make a strong argument out of this significant three-way interaction, we nevertheless fitted two additional models that predicted the likelihood of s-genitive production in the head noun condition subsets. We found that the interaction between the prime condition and problem difficulty was only significant in the Same Head Noun subset. The results of the subset models were added in the manuscript, and the fixed effect of the models were put in Appendix A. <<

#4 pg. 38, line 924-925 says: “The non-significant interaction between cognitive load and lexical overlap on structural priming was somewhat surprising…” and then discusses the implications of this non-effect. I assume that is referring to the interaction between Task Difficulty x Head Noun Condition in Experiment 2a. However, when reporting the Exp 2a results, the text says (line 740): “Unexpectedly, we also found … a significant two-way interaction between task difficulty and lexical overlap (χ2 = 7.213, df = 1, p = .007).” Please resolve this conflict or clarify which comparison you are referring to in the GD.

>>Author’s response: For the first interaction the reviewer refers to what we meant was rather the three-way interaction between prime condition, head noun condition, and problem difficulty. Sorry we did not make it clear. Now we have changed the expression to “It is somewhat surprising that the cognitive load effect on structural priming was not predominantly head-specific.” Thanks for pointing this out.<<

Minor comments:

#5 pg. 25, line 593: The exploratory analysis of Exp 1b investigating processing time (I assume also by using RT, as in Exp 1a) x to-be-recalled structure has df = 0.42. I do not understand how doing a model-comparison test can produce a non-integer degrees of freedom. (Unfortunately I can’t look at the model output to help me understand as I can’t find it in the submission.)

>>Author’s response: This was a typo. We have revised this result.<<

-There are a few times throughout the paper when results between experiments are compared numerically, but not statistically (e.g., pg. 26, comparing the size of the lexical boost effect in Exps 1a vs 1b, etc.). Please make those comparisons statistically explicit and report the significance/non-significance of the cross-experiment comparison.

>>Author’s response: Based on this comment, we have added two sections of cross-experiment analyses (Page 25-26 and Page 35) that compares the structural priming effects, lexical boost effects, and cognitive load effects between the structural priming experiments and sentence structure memory experiments. In both analyses, there were significant two-way interaction between prime condition and experiment, but the three-way interactions between prime condition, head noun condition, and experiment were not significant. We also calculated the Bayesian factors of these interactions to further examine to what extent these effects supported the alternative hypotheses.<<

#6 I wonder how difficult even the most difficult math problems were. Is carrying in addition really such a challenging task? Subjects are only spending 3 seconds doing the math problem in the hardest case. Please talk a little about why you think this might be enough of a cognitive load to disrupt processing.

>>Author’s response: In the design of the experiments we needed to find a balance between having an effortful task and having a relatively short task, given the transiency of the lexical boost effects. We needed to keep the difficult secondary task relatively short but effortful. The processing of a carrying problem is around 1900 ms slower than solving an easy problem. This suggests that a considerable amount of cognitive resource is taxed in solving carrying problems. Additionally, a similar manipulation of cognitive load significantly affected the preparation time of sentence production (Ferreira & Swets, 2002), we believe it is possible that the cognitive load manipulation in the current study makes a difference on the choice of syntactic structures.<<

#7 In only Exp 2b is there the expected context effect, whereby production of the less-common structure increased over the course of the experiment; in the other experiments, this effect goes in the opposite direction (the less-common structure decreases over the experiment). This is surprising, given that it’s the opposite direction as would be expected. The authors write that the increase effect (the expected direction) is “possibly indicating a practice effect.” Please discuss this further. Why would there be a practice effect in Exp 2b but not the other experiments with (approximately) the same task? Why would the other experiments show the opposite direction as previously-attested context effects?

>>Author’s response: The reversed cumulative effect on the production of the less preferred sentence structure (i.e., s-genitive) might be attributed to a surprisal effect. When the participants first perceived an s-genitive prime, they would be more prompted to use this structure in the ensuing task because the unexpected first encounter exerts a strong prediction error. As the experiment progressed, the participants might then regress to their preferred structure to alleviate the processing load in sentence production, resulting in a decreasing pattern of s-genitive production.

We argue such a reversed cumulative effect would not occur in the sentence structure memory experiments. Instead, a practice effect might manifest itself in these experiments. The production tasks in the sentence structure memory experiments was driven by an additional goal to accurately repeat the previously encoded sentence structure. We would expect that the accuracy of structure repetition gradually increased as the participants were getting familiarized with the task, resulting in higher chance to produce the less frequent structure as the experiment progressed. We added a brief discussion about these effects at Page 37 (for the practice effect) and Page 44 (for the reversed cumulative effects). <<

Textual/editing comments:

#8 The references are not all standardized – some use the journal abbreviation (J Mem Lang) and others the full name (Journal of experimental psychology: Learning, memory, and cognition).

>>Author’s response: We have changed all journal names into the abbreviated form as required by the journal's submission format.<<

-pg. 14: references example stimuli sentences before the sentences are produced in the text. This makes that entire paragraph difficult to follow; I suggest moving the sample sentences before the paragraph describing the stimuli conditions (i.e., at line 330).

>>Author’s response: Fixed. Thank you for pointing that out.<<

#9 pg. 17-18, lines 419-427: There’s a strange paragraph, maybe it’s supposed to be the figure caption for Figure 2? It’s presented in the middle of the text (with no accompanying figure, which is only at the end of the text body).

>>Author’s response: Yes, it is the figure caption for Figure 2; the journal's submission guidelines require us to put them in the text immediately after the paragraph in which the figure was first cited<<

#10 pg. 21, line 501 (same for Exp 1b): The authors write that the model for the supplementary analysis investigating RT of Exp 1a is in the Supplementary Material, but it’s not in Appendix A, B, or C (which are the only attached materials), so please include an explicit pointer to where I can find these results.

>>Author’s response: We apologize for this omission. We have now added these models in Appendix A.<<

#11 Please note in the figure captions what the error bars are: standard error, standard deviation, confidence interval, etc.

>>Author’s response: We have noted in the figure caption that the error bars reflect standard errors calculated for a by-participants analysis.<<

#12 Please make the y-axis for the two figures on the same scale. It makes it easier to compare the results across experiments.

>>Author’s response: Fixed. Thank you for pointing that out.<<

#13 Grammar/wording comments:

-pg. 5, line 102-3: “It has been several decades for the debate about how such structural priming effect comes about.” -> “It has been several decades since the debate began about…” (for example)

-pg. 7, line 164: “tried to answer A similar question”

-pg. 9, line 224: “These findings are coherent with…” consistent with?

-pg. 13, line 310, 318: “figurines” -> figures

-pg. 27, line 651: “we predicted that the secondary task posited a cognitive load” -> imposed a cognitive load?

-pg. 30, line 726: “minuses” -> minus

-pg. 35, line 856: “more often correctly” -> delete “correctly” perhaps?

-pg. 37, line 881: intertwinement -> entwining

-pg. 38, line 915-956: “similar as for the recall of sentence structure” -> “similar TO” ?

-pg. 39, line 932: “whereas the memory of sentence structure is prune to the load manipulation.” -> I am not sure what this sentence intends.

>>Author’s response: We have modified all the grammar errors.<<

Reviewer #2: • I appreciate the availability of the authors’ data.

#1 Overall, I agree with the authors that it is interesting to further explore cognitive load factors that affect priming. I believe the manuscript presents a nice beginning to this. I am, however, not convinced by Experiment 1a in this manuscript. The authors employ a confederate, which I appreciate. I think it is prudent to consider social factors. Perhaps some participants were high self-monitors and simply took more time answering questions to not be judged poorly by their experimental partner. If this were a factor, it may be that they were paying less attention to their partner’s descriptions in anticipation of their turns.

>>Author’s response: This is an interesting point. We argue below however that even in the absence of a physical interlocutor, the participants' utterances still had communicative value. Additionally, a reanlysis does not find evidence for differences in secondary task difficulty. Specifically, in Experiment 2a and 2b the participants were instructed that the utterances they would listen to came from a “participant” in the earlier test (who was in fact a confederate) and that their recording would be used in the upcoming tests of further participants. So we would argue that the production tasks in Experiment 2a and 2b also carried communicative purpose. The main difference in the social contexts of Experiments 1a-b and 2a-b would be the physical presence of an interlocutor. To examine whether the appearance of an interlocutor influenced participants’ processing time in problem solving, we conducted a between-experiment comparison of the processing time in filler arithmetic problem solving tasks. The materials in these filler problem solving tasks were identical in all four experiments. This enabled us to conduct a pairwise comparison of by-item mean processing time between experiments. We only examined the processing time of the hard problems (with or without carrying) because solving an easy problem requires little attention resources. If the participants took more time answering the questions when collaborating with an interlocutor, the processing time of a hard problem in the dialogue experiments should be longer than that in the monologue experiments. To test this hypothesis, we aggregated the processing time across items and compared the mean processing time between Experiment 1a and 2a as well as between 1b and 2b. The mean processing time in Experiment 1a was 3095 ms and that in Experiment 2a was 3097 ms. The standardized difference was negligible (Cohen's d for paired samples = -0.006). The difference of processing time between Experiment 1b (3102 ms) and 2b (3103 ms) was also negligible (Cohen's d for paired samples = -0.002). This indicates that the participants in the dialogue experiments did not take more time than those in the monologue experiments to solve the same filler task, suggesting no effect of the presence of an interlocutor on secondary task processing.<<

• Regarding the second series of experiments – I found it interesting that the authors decided to make two major changes – 1) the difficulty of the problems and 2) the dismissal of the confederate. I agree that the arithmetic problems, especially the “easy” ones, likely presented minimal, if any cognitive load on the participants. However, making two changes at once leads to questionable conclusions – how can you be sure which manipulation led to the change in significance?

>>Author’s response: We have argued in the response to comment #1 that the physical presence of an interlocutor had little impact on secondary task processing, and hence made little difference for the change of the cognitive load in Experiment 2a-b compared to Experiments 1a-b. We believe the main factor that drove the load increase was the more difficult secondary task.<<

• I think this is a useful addition to the literature. However, I would recommend running a third study with a confederate and the more difficult problems. It would round off these series of experiments nicely and address the concerns I (and the authors even noted) have about the manuscript.

>>Author’s response: Thank you for pointing this out. But because of the limited effect of this manipulation on the problem solving task, we have decided not to follow this suggestion<<

---

## [Decision Letter · Decision Letter 1]

8 Jul 2020

PONE-D-20-04033R1

The role of explicit memory in syntactic persistence: effects of lexical cueing and load on sentence memory and sentence production

PLOS ONE

Dear Dr. Zhang,

I am writing with regard to your manuscript, "The role of explicit memory in syntactic persistence: effects of lexical cueing and load on sentence memory and sentence production" (PONE-D-20-04033R1). I have received a set of comments from one of the original reviewers of your manuscript, and I have also reviewed your revision. The reviewer and I had very similar responses to your work:  you are presenting a nice set of experiments that contribute to our understanding of structural priming, but there are still issues with your statistical analyses and how they are presented. 

The reviewer raises three specific points about the analyses. 

1 -- Your procedure for dropping random effects from your models is not fully specified. For example, the reviewer notes that your strategy is to drop the most complex random effects (e.g., start with 3-way interaction, then the 2-way interactions, and so on), but it is not clear how you decide which effect to drop when there are multiple effects at the same level of complexity (e.g., multiple 2-way interactions). 

2 -- There appears to be an inconsistency between your stated strategy for simplifying your models (drop random correlations, then drop random slopes) and what appears in your code and text.

3 -- There also appears to be an inconsistency between the analyses specified in your code, the analyses described in the text, and the analyses presented in your Appendix. The confusion arises because you both a) present code for an analysis that includes all of the expected fixed effects, and then present the results of the analysis in the Appendix (i.e., a "full" analysis of your design), and b) present code suggesting that you are testing for specific effects using a model-comparison procedure (compare a model that has the target effect, and a model that does not) and generally focus on these specific effects in the text (as evidenced by the presentation of the X^2 statistic in the text, rather than the Z from the model results in the Appendix). The results in the text and in the Appendix are in accord, but it is unclear why you chose this particular way to handle the effects -- it would seem more reasonable to either stick with the results in the Appendix, or do the model comparison approach for all of your predictors (and present the X^2 values in your tables). I also think it would be helpful to have the full models reported in the text (i.e., put the tables in the text, rather than in the Appendix). 

Beyond these main points, the reviewer also provides a set of minor comments for you to consider and address.  

We look forward to receiving your revised manuscript.

Kind regards,

Mike Kaschak

Academic Editor

PLOS ONE

Reviewers' comments:

Reviewer's Responses to Questions

**Comments to the Author**

1. If the authors have adequately addressed your comments raised in a previous round of review and you feel that this manuscript is now acceptable for publication, you may indicate that here to bypass the “Comments to the Author” section, enter your conflict of interest statement in the “Confidential to Editor” section, and submit your "Accept" recommendation.

Reviewer #1: (No Response)

2. Is the manuscript technically sound, and do the data support the conclusions?

Reviewer #1: Partly

3. Has the statistical analysis been performed appropriately and rigorously? 

Reviewer #1: No

4. Have the authors made all data underlying the findings in their manuscript fully available?

Reviewer #1: Yes

5. Is the manuscript presented in an intelligible fashion and written in standard English?

Reviewer #1: Yes

6. Review Comments to the Author

Reviewer #1: This paper is still an interesting topic and contribution to the literature. However, there remain several inconsistencies or errors with the analyses which have the potential to affect the results. I feel the authors need to fix the analyses before I can have confidence that the results are in fact as they have been presented (and thus of course whether the discussion/interpretation follows).

MAJOR COMMENTS:

-There still is confusion about the running and reporting of the lmer models. Due to the discrepancies between what is reported in the paper for the overall analysis strategy, what is reported for each individual experiment, and the R code, in addition to the fact that the reported results often seem at odds with the means and sds shown in the tables/figures (as even the authors themselves note at times), I believe the authors need to very carefully check their analyses, and this must be resolved before I can have confidence in them. I detail several (potentially related) concerns below.

The authors write (pg. 22, line 112) "If the maximal model could not converge or showed singularity, we first dropped the random correlation terms, and then dropped one random factor at a time, starting from the interaction terms, until the model converged and no warning of singular fit was reported." They write something similar in their Response letter.

a) How did you decide which order to drop terms if a model did not converge? A common strategy is to iteratively drop the random effect which accounts for the least amount of variance, but it does not sound like this was what was done. The authors say their strategy was to start by dropping the most complex interaction term, but how did you decide the order of dropping terms at the same complexity level (i.e., the order of dropping among the 3 2-way interactions, or among the 3 main effects)? It looks like terms were dropped based on the order that they were written in the model starting from the bottom-up; in order words, arbitrarily within a complexity level. That is not a good strategy, as dropping random terms can affect significance and thus should not be done arbitrarily. Dropping a random effect which accounts for a substantial amount of variance can have a big effect on the observed significance of fixed effects, and while this is necessary sometimes for convergence/singularity reasons, it needs to be done systematically.

b) The authors write that they first dropped random correlations, and then started dropping random slopes terms. However, most of the models (the main model for Exp 1b, 2a, and 2b) - both in the R code and as written in the paper text - contradict this stated strategy: Those models RETAIN (e.g.) the by-subjects random correlations, but DROP almost all of the by-subjects random effects. (But Exp 1a follows this stated strategy, so they differ between each other.) That is, this model did NOT first drop the by-subject random correlations and then start dropping random effects. It’s really important to both (1) have the description in the paper body match your actual model, and (2) to have a systematic and consistent strategy for pruning a model’s terms, as this can have a major effect on the model’s output.

c) In both the R code and the text body, only some of the factors appear to be tested. In both Exp 1a and 1b, there are models testing only these fixed effects:

-Prime Structure

-Trial Number

-Prime Structure x Head Noun

-Prime Structure x Task Difficulty

-Prime Structure x Head Noun x Task Difficulty

Where are the models testing the other fixed effects? Did you test, for example, for the main effect of Head Noun or Task Difficulty? If you didn’t test all factors, you should say that explicitly in the paper and justify why not. Even if some factors are not theoretically interesting, it is very non-standard to not even test for/report those factors if they were entered into your analysis.

The cross-experiment analysis of 1a vs. 1b similarly omits testing of a number of factors.

Strangely, in Exp 2a, the authors suddenly do test for the main effect of Problem Difficulty (but still not Head Noun), and Problem Difficulty x Lexical Overlap [sic].

But then testing for those two fixed effects disappears again in Exp 2b, which only tests for the 5 fixed effects I noted above.

So again we have a consistency problem.

MINOR COMMENTS:

-Table 1: I understand, I had misunderstood what the numbers in Table 1 represented. The authors respond that they changed the title of the table to clarify this; I admit the title looks identical to the previous one except that one sentence was moved from the top to the bottom of the table, but I leave this to the authors to figure out.

-There are some drastic changes between different versions of lme4, so I suggest you report the version of the package that you used.

-pg 29, line 292-293: The authors write "structural priming effect may persist over at least one intervening task, irrespective of what type of interference is employed." I do not think this is a fair statement; the authors have tested one type of secondary task (addition math problems); they don’t have evidence how any other type of task will or will not interfere.

-The authors reference a condition called "Lexical Overlap" and "Overlap Condition" (pg. 36, line 439 and continuing on that page). Is this supposed to be “Head Noun Condition” or is this a new contrast they have introduced?

-The authors write (pg. 29, lines 281-283) that the "Inverse of Bayes factor" is 0.023 or 0.020. Do they really mean “inverse”? If that’s accurate, that means the Bayes factor is around 50 (1/0.020), which is very strong evidence in favor of H1 (cf Jeffreys 1961).

-The authors provide a short response to my previous question about whether the math problems provide enough of a cognitive load to disrupt processing in their Response letter. However, I didn’t see this in the manuscript itself. While I appreciate the personalized response, it would be useful if other readers could see this discussion as well.

MINOR WORDING EDITS:

-pg. 29, line 291-292: “suggest that THE structural priming effect may persist”

-pg. 109, line 326-327: “One unexpected result in the priming experiments was a negative correlation between the critical trial number and the likelihood of s-genitive production.”  “priming experiment” [singular], as you only found this negative correlation in Exp 1a, not 1b.

7. PLOS authors have the option to publish the peer review history of their article (what does this mean?). If published, this will include your full peer review and any attached files.

Reviewer #1: No

---

## [Author Response · Author response to Decision Letter 1]

28 Jul 2020

Reviewer #1: This paper is still an interesting topic and contribution to the literature. However, there remain several inconsistencies or errors with the analyses which have the potential to affect the results. I feel the authors need to fix the analyses before I can have confidence that the results are in fact as they have been presented (and thus of course whether the discussion/interpretation follows).

MAJOR COMMENTS:

#1 There still is confusion about the running and reporting of the lmer models. Due to the discrepancies between what is reported in the paper for the overall analysis strategy, what is reported for each individual experiment, and the R code, in addition to the fact that the reported results often seem at odds with the means and sds shown in the tables/figures (as even the authors themselves note at times), I believe the authors need to very carefully check their analyses, and this must be resolved before I can have confidence in them. I detail several (potentially related) concerns below.

The authors write (pg. 22, line 112) "If the maximal model could not converge or showed singularity, we first dropped the random correlation terms, and then dropped one random factor at a time, starting from the interaction terms, until the model converged and no warning of singular fit was reported." They write something similar in their Response letter.

a) How did you decide which order to drop terms if a model did not converge? A common strategy is to iteratively drop the random effect which accounts for the least amount of variance, but it does not sound like this was what was done. The authors say their strategy was to start by dropping the most complex interaction term, but how did you decide the order of dropping terms at the same complexity level (i.e., the order of dropping among the 3 2-way interactions, or among the 3 main effects)? It looks like terms were dropped based on the order that they were written in the model starting from the bottom-up; in order words, arbitrarily within a complexity level. That is not a good strategy, as dropping random terms can affect significance and thus should not be done arbitrarily. Dropping a random effect which accounts for a substantial amount of variance can have a big effect on the observed significance of fixed effects, and while this is necessary sometimes for convergence/singularity reasons, it needs to be done systematically.

<<Authors’ response: We did adopt a bottom-up strategy of term dropping in the random model in the previous model selections. The reason we employed this method was to avoid the potential complication when a lower-level random effect accounts for less variance than a higher-level random effect.

In the newest version of the manuscript, we adapted the method to your suggested strategy. Now, in the LME model analysis, we arranged the model selection based on both the complexity of the terms and the variance of the random effects. When the maximal model could not converge or showed singularity, we first dropped the random correlation terms (once and for all, see the response to Comment #1b), and then dropped one random factor at a time, starting from the most complex interaction terms. When there were multiple terms with the same complexity, we compared the variances of the random effects in the last model and dropped the term with the least amount of variance. We repeated this selection process until we fitted the first model that converged and showed no singularity. We have revised our description of the model selection method in the manuscript accordingly (p22).

We have updated the information about the final model and the fixed effects of the model in the manuscript. We have also updated our model selection method in the OSF repository (link: https://osf.io/6utkf/). There you could also find two additional scripts that tracked the process of model selection.>>

b) The authors write that they first dropped random correlations, and then started dropping random slopes terms. However, most of the models (the main model for Exp 1b, 2a, and 2b) - both in the R code and as written in the paper text - contradict this stated strategy: Those models RETAIN (e.g.) the by-subjects random correlations, but DROP almost all of the by-subjects random effects. (But Exp 1a follows this stated strategy, so they differ between each other.) That is, this model did NOT first drop the by-subject random correlations and then start dropping random effects. It’s really important to both (1) have the description in the paper body match your actual model, and (2) to have a systematic and consistent strategy for pruning a model’s terms, as this can have a major effect on the model’s output.

<<Authors’ response: The reason that the exclusion of the random correlation was not symmetrical for the by-subject and the by-item random effects is because that we fitted models with and without random correlations in each iteration of the term dropping. We always started a model with random correlations. If that model did not converge, we would fit a model that dropped the random correlation for subjects and a model that dropped the random correlation for items. If both models converged, we selected the model based on the goodness-of-fit of the models. If one of the models converged, that model would be the final model. If neither of the models converged, we would fit a model with no random correlation. We did this iteratively each time after a term was dropped from the random model. Sorry we did not make it very clear in the manuscript.

In the newest version of the manuscript, we aligned the method of model selection with the description in the Results section of the manuscript. More specifically, now we started the data analysis with the maximal model, if that model could not converge, we fitted another model with the most complex random effect structure, but the random correlation was dropped. If the reduced model could not converge, we continued with the model selection strategy described in the response of comment #1a and never considered the random correlation again. We hope now the model selection method is consistent with our description in the manuscript. >>

c) In both the R code and the text body, only some of the factors appear to be tested. In both Exp 1a and 1b, there are models testing only these fixed effects:

-Prime Structure

-Trial Number

-Prime Structure x Head Noun

-Prime Structure x Task Difficulty

-Prime Structure x Head Noun x Task Difficulty

Where are the models testing the other fixed effects? Did you test, for example, for the main effect of Head Noun or Task Difficulty? If you didn’t test all factors, you should say that explicitly in the paper and justify why not. Even if some factors are not theoretically interesting, it is very non-standard to not even test for/report those factors if they were entered into your analysis.

The cross-experiment analysis of 1a vs. 1b similarly omits testing of a number of factors.

Strangely, in Exp 2a, the authors suddenly do test for the main effect of Problem Difficulty (but still not Head Noun), and Problem Difficulty x Lexical Overlap [sic].

But then testing for those two fixed effects disappears again in Exp 2b, which only tests for the 5 fixed effects I noted above.

So again we have a consistency problem.

<<Authors’ response: In the newest version of the manuscript, we only reported the fixed effects of the LME models (estimates, standard errors, and p-value) but not the results of model comparison. The tables of the fixed effects in the LME models for each experiment was moved from Appendix A to the results sections.

We further modified the report of results in such a way that we first reported theoretically interesting effects that were significant, followed by the theoretically interesting effects that were not significant, the unexpectedly significant effects, and ended with the other non-significant effects. This way, we hope the report of model results is more consistent.

Due to the model update and change in the reporting of the results, the three-way interaction between prime condition, head noun condition, and problem difficulty was no longer significant in Experiment 2a, while this interaction became significant in Experiment 2b. Accordingly, we deleted the section about the subset analysis in Experiment 2a and added a subset analysis in Experiment 2b (p43-p44). And the change of the patterns of the interactions led to some modification of the discussion about the lexical-specificity of the cognitive load effect on structure persistence (p53).>>

MINOR COMMENTS:

#2 Table 1: I understand, I had misunderstood what the numbers in Table 1 represented. The authors respond that they changed the title of the table to clarify this; I admit the title looks identical to the previous one except that one sentence was moved from the top to the bottom of the table, but I leave this to the authors to figure out.

<<Authors’ response: We made some further changes in order to clarify what each column represents in the table. First, in the first row of the header, we showed the name of the three independent variables. Second, we changed the wording of the table title to …the proportion of s-genitive responses out of all s-genitive and of-genitive responses... Third, we added a column to the right of the table that reports the structure repetition effects in each head noun condition (i.e., the proportion of s-genitive in s-genitive condition minus that in of-genitive condition). >>

#3There are some drastic changes between different versions of lme4, so I suggest you report the version of the package that you used.

<<Authors’ response: We have reported the version of lme4 used in the analysis (p21).>>

#4 pg 29, line 292-293: The authors write "structural priming effect may persist over at least one intervening task, irrespective of what type of interference is employed." I do not think this is a fair statement; the authors have tested one type of secondary task (addition math problems); they don’t have evidence how any other type of task will or will not interfere.

<< Authors’ response: We have deleted this statement.>>

#5 The authors reference a condition called "Lexical Overlap" and "Overlap Condition" (pg. 36, line 439 and continuing on that page). Is this supposed to be “Head Noun Condition” or is this a new contrast they have introduced?

<<Authors’ response: We have unified the terms used in the manuscript>>

#6 The authors write (pg. 29, lines 281-283) that the "Inverse of Bayes factor" is 0.023 or 0.020. Do they really mean “inverse”? If that’s accurate, that means the Bayes factor is around 50 (1/0.020), which is very strong evidence in favor of H1 (cf Jeffreys 1961).

<< Authors’ response: Indeed, the inverse of Bayes factor (or BF10) here means the results that 1 is divided by the Bayes factor (BF01). It was argued that the bigger the inverse of Bayes factor is, the stronger the evidence that supports the alternative hypothesis (Raftery, 1995). That is why we followed the practice in previous literature and reported BF10 in the cross-experiment analyses. However, given we predicted that there would be no difference in the magnitudes of interactions between the two experiments, it is very much reasonable to only use Bayes factor (BF01) when we report the results of Bayesian analysis. So in the newest version of the manuscript, we decided to report Bayes factor (BF01) instead of its inverse (BF10).

We have made it clear that the Bayes factors that were reported in the manuscript were exclusively BF01. We then interpreted the BF01 only in terms of to what extent it supported or opposed the null hypothesis. This way, we hope to make the report of Bayes factors more consistent.>>

#7 The authors provide a short response to my previous question about whether the math problems provide enough of a cognitive load to disrupt processing in their Response letter. However, I didn’t see this in the manuscript itself. While I appreciate the personalized response, it would be useful if other readers could see this discussion as well.

<< Authors’ response: We have added parts in the manuscript that justified the design of the secondary task (p13, p35, and p47).>>

MINOR WORDING EDITS:

#8 pg. 29, line 291-292: “suggest that THE structural priming effect may persist”

#9 pg. 109, line 326-327: “One unexpected result in the priming experiments was a negative correlation between the critical trial number and the likelihood of s-genitive production.”  “priming experiment” [singular], as you only found this negative correlation in Exp 1a, not 1b.

<<Author’s response: We have modified all the wording errors. Thank you for pointing them out.>>

---

## [Decision Letter · Decision Letter 2]

22 Sep 2020

PONE-D-20-04033R2

The role of explicit memory in syntactic persistence: effects of lexical cueing and load on sentence memory and sentence production

PLOS ONE

Dear Dr. Zhang,

I am writing with regard to your manuscript, "The role of explicit memory in syntactic persistence: effects of lexical cueing and load on sentence memory and sentence production" (PONE-D-20-04033R2). I have received comments the reviewer who handled your last submission of the manuscript, and I have reviewed your manuscript myself. The reviewer and I find that you have done a good job of handling the majority of the concerns raised in the last round of reviews, leaving only a few relatively straightforward concerns to address. 

The main point to address comes from the reviewer, who notes that there appears to be an inconsistency between the sign of the effects listed in your tables and your stated procedure for defining a reference level for the comparisons. Please double check this to make sure that the results are presented correctly. Beyond this, there are a few other smaller concerns to address. 

We look forward to receiving your revised manuscript.

Kind regards,

Mike Kaschak

Academic Editor

PLOS ONE

Reviewers' comments:

Reviewer's Responses to Questions

**Comments to the Author**

1. If the authors have adequately addressed your comments raised in a previous round of review and you feel that this manuscript is now acceptable for publication, you may indicate that here to bypass the “Comments to the Author” section, enter your conflict of interest statement in the “Confidential to Editor” section, and submit your "Accept" recommendation.

Reviewer #1: (No Response)

2. Is the manuscript technically sound, and do the data support the conclusions?

Reviewer #1: Yes

3. Has the statistical analysis been performed appropriately and rigorously? 

Reviewer #1: I Don't Know

4. Have the authors made all data underlying the findings in their manuscript fully available?

Reviewer #1: Yes

5. Is the manuscript presented in an intelligible fashion and written in standard English?

Reviewer #1: Yes

6. Review Comments to the Author

Reviewer #1: The authors have largely addressed my statistical comments on the previous version of the manuscript. There remain a few concerns but most of them are minor.

STATISTICS/ANALYSIS COMMENTS:

1. Table 2 lists the fixed effects estimates for Exp 1a with the caption, “Prime condition (S-genitive as the baseline level), head noun condition (Same Head Noun as the baseline level), problem difficulty (Easy Problem as the baseline level) were in mean-centered form.” But the estimates and z-scores listed in the table all seem to have the wrong sign. If S-genitive is coded as the baseline (reference) level, then the estimate shows the change from S-genitive to of-genitive prime. Subjects produced more S-genitives following S-genitive primes than of-genitive primes (see Table 1); that means the change from S-gen prime to of-gen prime is negative. But the estimate listed in Table 2 for Prime Condition is positive. There is a similar problem for Head Noun Condition – more S-genitives were produced in the Same than the Different Head Noun Condition, and Same was coded as the baseline level. That means the estimate for Head Noun Condition should be negative, but it is positive. This backwards-sign problem appears to be the case for all of the models reported in the paper (the four experiments plus the cross-experiment analyses). (Though perhaps not for the effect of critical trial number, which is continuous rather than categorical.)

This speaks to some underlying error that I’m having trouble diagnosing. I don’t know if the error comes about because of (a) type-os in the text of inverting all of the signs, (b) miscoding your factors in the code and analysis, or (c) me misunderstanding what you did (in which case it should be clearer because I am trying really hard to figure it out). But whatever is causing this problem, the output numbers don’t make sense.

2. The authors write, “Based on the standard interpretation of Bayes factors as evidence for null hypotheses [65], BF01 that ranges from 1 to 3 can be taken as weak evidence for the null hypothesis. The higher a BF01, the more evidence in support of the null hypothesis (3-20: positive evidence; 20-150: strong evidence; > 150: very strong evidence).”

I’ve always seen Bayes Factor for experimental hypothesis testing written as strength of evidence in favor of the Alternative (H1), rather than evidence in favor of the Null (H0). For example, as noted in [Kass, R.E. and Raftery, A. (1995) Bayes Factors, Journal of the American Statistical Association, 90: 773-795.], “When comparing results with standard likelihood ratio tests, it is convenient to instead put the null hypothesis in the denominator and thus use B10 as the Bayes factor.” Additionally, Jeffreys (1961), where Bayes Factor was originally proposed, lists BF values as strength of evidence for H1, so numbers >1 show evidence in favor of H1 and numbers <1 show evidence in favor of H0. Anyway, you do write explicitly that you’re using BF01 so I suppose that’s alright, but I suspect it will be confusing to readers who see a decimal number and yet the interpretation is that the evidence favors H1 (as I was on the previous version of the manuscript).

3. There is no report of all of the raw condition means in the paper, which makes interpreting some of your results difficult or impossible. For example, in Exp 2a, there is a main effect of Problem Difficulty, but I don’t know how that is actually manifest in the data. The text lists the p-value but no condition means. Table 5 lists the estimate which ostensibly tells me the direction and magnitude of the effect in standardized terms, (though see earlier point regarding sign) but no raw values. Table 1 lists raw values but only for Prime Condition x Head Noun Condition, collapsing across Problem Difficulty. The figures only show Head Noun Condition x Problem Difficulty because the DV is the Prime Condition difference score. So there’s a main effect of Problem Difficulty and I have no idea what that actually looks like in terms of raw condition means.

MINOR COMMENTS:

4. The code to normalize the key press RT and trial number variables is missing from the model selection code so it doesn’t run as-is. (You can leave it if you want but just so you know that I had to modify your code to run it.)

5. There are quite a number of type-os, extra or missing words, and grammatical errors throughout the text. I list a few here but this is non-exhaustive (just the ones I happened to notice) and I recommend careful proof-reading.

-pg. 22: “This is because that by using contrastive coding, the fixed effects of the model are informative about the main effects and the interactions”  remove “that”

-pg. 29: “Similar to Experiment 1a, there was also a main effect of the head noun condition (pz

630 < .001), which might also BE due to that the head noun overlap effect…”  add “be”, remove “that”

-pg. 32: “In addition, the theoretically interestING interactions that involved the contrast between the two experiments were examined by estimating Bayes factors (BF01) using Bayesian Information Criteria.”  add “ing”

7. PLOS authors have the option to publish the peer review history of their article (what does this mean?). If published, this will include your full peer review and any attached files.

Reviewer #1: No

---

## [Author Response · Author response to Decision Letter 2]

1 Oct 2020

Reviewer #1: The authors have largely addressed my statistical comments on the previous version of the manuscript. There remain a few concerns but most of them are minor.

STATISTICS/ANALYSIS COMMENTS:

1. Table 2 lists the fixed effects estimates for Exp 1a with the caption, “Prime condition (S-genitive as the baseline level), head noun condition (Same Head Noun as the baseline level), problem difficulty (Easy Problem as the baseline level) were in mean-centered form.” But the estimates and z-scores listed in the table all seem to have the wrong sign. If S-genitive is coded as the baseline (reference) level, then the estimate shows the change from S-genitive to of-genitive prime. Subjects produced more S-genitives following S-genitive primes than of-genitive primes (see Table 1); that means the change from S-gen prime to of-gen prime is negative. But the estimate listed in Table 2 for Prime Condition is positive. There is a similar problem for Head Noun Condition – more S-genitives were produced in the Same than the Different Head Noun Condition, and Same was coded as the baseline level. That means the estimate for Head Noun Condition should be negative, but it is positive. This backwards-sign problem appears to be the case for all of the models reported in the paper (the four experiments plus the cross-experiment analyses). (Though perhaps not for the effect of critical trial number, which is continuous rather than categorical.)

This speaks to some underlying error that I’m having trouble diagnosing. I don’t know if the error comes about because of (a) type-os in the text of inverting all of the signs, (b) miscoding your factors in the code and analysis, or (c) me misunderstanding what you did (in which case it should be clearer because I am trying really hard to figure it out). But whatever is causing this problem, the output numbers don’t make sense.

<< Author’s response: Thank you so much for pointing out this problem. The signs in Table 2 were not wrong. What went wrong was that we made typos in the table title. The levels that we coded as -0.5 (Of-genitive, Different Head Noun, and Difficult Problem) were supposed to be taken as the reference level. However, we mistakenly reported the levels coded as 0.5 (S-genitive, Same Head Noun, and Easy Problem) as the reference level. We were very sorry for this unfortunate mistake. Now we have corrected the naming of the reference level in Table 2-7.>>

2. The authors write, “Based on the standard interpretation of Bayes factors as evidence for null hypotheses [65], BF01 that ranges from 1 to 3 can be taken as weak evidence for the null hypothesis. The higher a BF01, the more evidence in support of the null hypothesis (3-20: positive evidence; 20-150: strong evidence; > 150: very strong evidence).”

I’ve always seen Bayes Factor for experimental hypothesis testing written as strength of evidence in favor of the Alternative (H1), rather than evidence in favor of the Null (H0). For example, as noted in [Kass, R.E. and Raftery, A. (1995) Bayes Factors, Journal of the American Statistical Association, 90: 773-795.], “When comparing results with standard likelihood ratio tests, it is convenient to instead put the null hypothesis in the denominator and thus use B10 as the Bayes factor.” Additionally, Jeffreys (1961), where Bayes Factor was originally proposed, lists BF values as strength of evidence for H1, so numbers >1 show evidence in favor of H1 and numbers <1 show evidence in favor of H0. Anyway, you do write explicitly that you’re using BF01 so I suppose that’s alright, but I suspect it will be confusing to readers who see a decimal number and yet the interpretation is that the evidence favors H1 (as I was on the previous version of the manuscript).

<< Author’s response: Thanks for the suggestion. We have modified the report of Bayes factors so that it is now focusing on BF10.>>

3. There is no report of all of the raw condition means in the paper, which makes interpreting some of your results difficult or impossible. For example, in Exp 2a, there is a main effect of Problem Difficulty, but I don’t know how that is actually manifest in the data. The text lists the p-value but no condition means. Table 5 lists the estimate which ostensibly tells me the direction and magnitude of the effect in standardized terms, (though see earlier point regarding sign) but no raw values. Table 1 lists raw values but only for Prime Condition x Head Noun Condition, collapsing across Problem Difficulty. The figures only show Head Noun Condition x Problem Difficulty because the DV is the Prime Condition difference score. So there’s a main effect of Problem Difficulty and I have no idea what that actually looks like in terms of raw condition means.

<< Author’s response: We have added in the Results sections all of the raw condition means.>>

MINOR COMMENTS:

4. The code to normalize the key press RT and trial number variables is missing from the model selection code so it doesn’t run as-is. (You can leave it if you want but just so you know that I had to modify your code to run it.)

<< Author’s response: We have added normalization of the key press RT and trial number variables in the scripts with model selection.>>

5. There are quite a number of type-os, extra or missing words, and grammatical errors throughout the text. I list a few here but this is non-exhaustive (just the ones I happened to notice) and I recommend careful proof-reading.

-pg. 22: “This is because that by using contrastive coding, the fixed effects of the model are informative about the main effects and the interactions”  remove “that”

-pg. 29: “Similar to Experiment 1a, there was also a main effect of the head noun condition (pz

630 < .001), which might also BE due to that the head noun overlap effect…”  add “be”, remove “that”

-pg. 32: “In addition, the theoretically interestING interactions that involved the contrast between the two experiments were examined by estimating Bayes factors (BF01) using Bayesian Information Criteria.”  add “ing”

<< Author’s response: We have modified all the grammar errors mentioned above and other errors we could find in the text. >>

---

## [Editor Report · Decision Letter 3]

6 Oct 2020

The role of explicit memory in syntactic persistence: effects of lexical cueing and load on sentence memory and sentence production

PONE-D-20-04033R3

Dear Dr. Zhang,

We’re pleased to inform you that your manuscript has been judged scientifically suitable for publication and will be formally accepted for publication once it meets all outstanding technical requirements.

Kind regards,

Michael Kaschak

Academic Editor

PLOS ONE
---

## [Editor Report · Acceptance letter]

12 Oct 2020

PONE-D-20-04033R3 

The role of explicit memory in syntactic persistence: effects of lexical cueing and load on sentence memory and sentence production 

Dear Dr. Zhang:

I'm pleased to inform you that your manuscript has been deemed suitable for publication in PLOS ONE. Congratulations! Your manuscript is now with our production department. 

Kind regards, 

on behalf of

Dr. Michael P. Kaschak 

Academic Editor

PLOS ONE